# Can Formal Security Verification Really Be Optional? Scrutinizing the Security of IMD Authentication Protocols

**DOI:** 10.3390/s21248383

**Published:** 2021-12-15

**Authors:** Daniel Gerbi Duguma, Ilsun You, Yonas Engida Gebremariam, Jiyoon Kim

**Affiliations:** 1Department of Information Security Engineering, Soonchunhyang University, Asan-si 31538, Choongchungnam-do, Korea; danielgerbi2005@gmail.com; 2Department of ICT Environmental Health System, Soonchunhyang University, Asan-si 31538, Choongchungnam-do, Korea; yonas.engidag@gmail.com

**Keywords:** implantable medical device, IMD security, IMD authentication protocol, formal security verification

## Abstract

The need for continuous monitoring of physiological information of critical organs of the human body, combined with the ever-growing field of electronics and sensor technologies and the vast opportunities brought by 5G connectivity, have made implantable medical devices (IMDs) the most necessitated devices in the health arena. IMDs are very sensitive since they are implanted in the human body, and the patients depend on them for the proper functioning of their vital organs. Simultaneously, they are intrinsically vulnerable to several attacks mainly due to their resource limitations and the wireless channel utilized for data transmission. Hence, failing to secure them would put the patient’s life in jeopardy and damage the reputations of the manufacturers. To date, various researchers have proposed different countermeasures to keep the confidentiality, integrity, and availability of IMD systems with privacy and safety specifications. Despite the appreciated efforts made by the research community, there are issues with these proposed solutions. Principally, there are at least three critical problems. (1) Inadequate essential capabilities (such as emergency authentication, key update mechanism, anonymity, and adaptability); (2) heavy computational and communication overheads; and (3) lack of rigorous formal security verification. Motivated by this, we have thoroughly analyzed the current IMD authentication protocols by utilizing two formal approaches: the Burrows–Abadi–Needham logic (BAN logic) and the Automated Validation of Internet Security Protocols and Applications (AVISPA). In addition, we compared these schemes against their security strengths, computational overheads, latency, and other vital features, such as emergency authentications, key update mechanisms, and adaptabilities.

## 1. Introduction

The need for continuous monitoring of physiological information of critical organs of the human body, combined with the ever-growing field of electronics and sensor technologies, and the colossal opportunities brought by 5G connectivity, have made implantable medical devices (IMDs) the most necessitated devices in the health arena. This is clearly shown by the global IMD market share, which was worth USD 96.6 billion in 2018 [1] and grew to around USD 103.3 Billion in 2019, and will likely rise to USD 148.8 Billion in 2024 [2].

IMDs possess several applications to help manage numerous health conditions. These include controlling the heart rhythm using cardiac pacemakers, heart support using ventricular assist devices, and chronic spinal pain reliefs using spinal cord stimulators [3]. Furthermore, they extend their applications by enabling wireless communication technologies that help manage the interaction between IMDs and external devices in wireless body area networks (WBANs) [4,5]. IMDs functioning in WBANs have made a significant contribution in resolving several challenges in both medical and non-medical fields, yet they have their hurdles.

Despite their critical roles in improving human health conditions, IMDs have various challenges, among which, limitations of resource (power, storage, computation, etc.) and security concerns are the most serious. The former challenge is directly related to their small size and inflexibility since they are implanted in the human body. Concerning the latter, IMDs are susceptible to many security and privacy threats that put a patient’s life in danger [6]. Some of the most common security problems that IMDs face are impersonation, requesting confidential information, causing a shock to the patient, reprogramming of IMD, etc. Moreover, security assaults (e.g., side-channel attacks) targeting a wide range of internet of things (IoT) processors, such as the Cortex-A platform, also threaten the wellbeing of IMDs [7].

To date, many countermeasures have been taken to keep the confidentiality, integrity, and availability of IoT systems, along with different privacy and safety mechanisms [8,9,10,11,12]. In particular, to IMDs, different researchers have proposed several solutions that can be categorized into three main groups: cryptographic, access control, and misbehavior detection. The first group of solutions utilizes cryptographic rudiments (including public-key encryption, symmetric-key encryption, cryptographic hash functions, etc.) [13,14]. Access control mechanisms [15,16,17], on the other hand, protect IMDs from unauthorized access by employing different techniques, such as certificates and lists, designation-based, juxtaposition-based, and biometric-based [6]. The last type of method involves malicious behavior detection to shield IMDs from a range of attacks that may not be easily addressed by the former two solutions [18,19].

IMDs are very sensitive as they are implanted in the human body, and the patients depend on them for the proper functioning of their vital organs. Moreover, due to their resource limitations and the open channel utilized for data transmission, they are intrinsically vulnerable to several attacks, such as distributed denial of service with different attacker intentions [20]. Hence, failing to secure them would put the life of the patient in jeopardy, and damage the reputations of the manufacturer. Consequently, it is imperative to carefully examine the security of the IMD authentication protocols for any vulnerabilities. To do so, we followed two methods. First, we conducted an extensive literature review to understand the operations, architectural perspectives, critical security, and privacy requirements and proposed solutions. We also leveraged empirical data that approximated delays introduced by cryptographic operations for comparative analysis of the authentication protocols. Next, we used two well-known security verification approaches, BAN logic [21] and AVISPA [22], to formally analyze the authentication protocols. Unfortunately, many security protocols designed for IMDs are not formally verified, or they use only one verification method [23,24,25,26,27,28,29,30].

The main contributions of this research work can be summarized as follows:We examined various security and privacy requirements along with numerous threats that surround IMDs.We performed formal security validation of the contemporary authentication schemes based on BAN logic and AVISPA against several security goals.We compared these schemes concerning security strength, computational overhead, latency, and additional features, such as emergency authentication, adaptiveness, and key update mechanisms.

The rest of the paper continues as follows. Section 2 describes the components of a typical IMD system architecture. Section 3 outlines various security and privacy requirements, issues, and proposed solutions. Section 4 presents the formal security analysis of different IMD authentication protocols using BAN logic and AVISPA. Section 5 puts forward the discussion of the results found in Section 4. Section 6 describes the comparative analysis of the authentication protocols concerning functionality, computational overhead, and communication latency. Finally, Section 7 concludes the paper.

## 2. Typical IMD System Architecture

IMDs play a critical role in sensing vital physiological information, which is then sent out to an external device via the wireless medium for different actions, such as remote monitoring and drug delivery. Typically, such systems are assembled from various components, as shown in Figure 1, among which the following are the main ones.

Sensor devices. These are small, in-body implanted, battery-powered, and wireless communication enabled sensors to sense, collect, and send patient information to a controller. In general, there are three categories (based on the data measured/collected) of such sensors: those that measure vital physiological information (such as glucose level, EEG, ECG, etc.), those that gather main environmental parameters, such as humidity, temperature, and pressure, and those that measure signals related to the human body movements [31].

Battery. Implanted sensors need the power to sense information on the body and produce an output. The source of energy for active implants comes from batteries. These batteries can be chargeable or non-chargeable, depending on the sensor type [32], and external or through independent power sources [33]. While the former approach uses optical charging, ultrasonic transducer, and inductive coupling, the latter uses the body environment energy to generate electrical energy for IMDs. Either way, efficient power management is a must since it is difficult (or not desirable) to change batteries now and then. Hence, batteries fixed with these implants should serve for a prolonged period.Memory. Memory is vital for the proper functioning of IMDs. It enables implants to store sensed data, configurations, and other important information (such as security keys). The device memory is generally non-volatile (read-only memory (ROM)), retaining its contents regardless of the power supply. In addition, the electrically erasable programmable ROM (EEPROM) and flash memories can be good candidates [32].Processing unit. The processing unit is the brain of the entire IMD system, which processes instructions and control signals. The processing unit actively directs the communication between IMDs and external devices, efficient power and transceiver management, and is responsible for other essential tasks, such as sensing and processing data [32].Transceiver. To communicate different sensed data to the external devices (such as a programmer) and receive other information from the external devices, IMDs need to establish a wireless medium. An electronic device, known as a transceiver (transmitter and receiver), assists this exchange of information. A specifically designed transceiver called the Medical Implant Communication System (MICS) is available for medical implants with low-power, short-range, and high data rate features [34].Application-Specific Components. These components are optional, meaning they may not appear in all implanted devices. One good illustration is the Smart Implant Security Core (SISC) [35]. Communication between IMD and a programmer via wireless medium passes through this device. It runs an energy-efficient security protocol by using energy harvesting when it performs authentication with the programmer. Apart from that, SISC helps defend against denial-of-service attacks, particularly resource exhaustion attacks.Wireless Identification and Sensing Platform (WISP). One of the significant constraints of implanted devices is related to power. These devices reside in the human body, making them challenging to recharge or frequently change. Hence, a device called WISP is proposed [32]. Using WISP, therefore, it is possible to conserve the battery of an IMD, especially during an authentication process, as it harvests energy from the reader via radiofrequency.Programmer/Controller. Sensing or measuring vital physiological states is only half of the primary goal of using implants. The sensors should also convey the sensed information to an external device (a specially designed controller or a smartphone) near the IMDs. Apart from collecting sensed information from the implants, programmers/controllers assist in configuration setup and regulation of therapy, among others.

## 3. Security and Privacy Requirements, Threats, and Proposed Solutions

IMDs encounter several challenges, from their conception through their operation. These devices are implanted and severely limited in terms of power, storage, and computing capabilities, making it challenging to build effective communication technologies and security mechanisms. In this regard, IMDs must satisfy various security requirements to withstand the ever-increasing attacks that target them.

The privacy of patients is of paramount importance. Two critical issues in this regard are user anonymity and non-traceability [6,36]. The former refers to a strong requirement that it should be impossible (or difficult enough) for the attacker to intercept the patient’s identity from the messages exchanged. Often, this is the first step towards an impersonation attack in which an adversary identifies the user’s real identity to fool the other party. Non-traceability, on the other hand, protects the IMD by making it difficult for an attack to know where the patient is or from where he is communicating. As a result, the locations of patients remain confidential, and any acts they conduct cannot be traced back to them by an unauthorized entity.

### 3.1. Security and Privacy Requirements

Here, we describe nine essential security requirements relevant to the IMDs:Confidentiality: the physiological information collected by IMDs is often sent out to a reader via a wireless medium, which both authorized and malicious users can observe. Accordingly, it is essential to encrypt this information to protect the data transmitted from exploitation by the adversaries sitting between the IMD and the reader.Integrity: protecting the integrity of the information transmitted via the wireless link in IMD reader communication defends against unauthorized modification. In addition, when illegitimate users tamper with the data, it should be known by the authorized users that the data is modified.Availability: this is one of the three security triads (confidentiality, integrity, and availability) that has the objective of making the IMD-enabled system accessible to authorized users despite the presence of adversaries.Mutual authentication: unless authorized access is in place, an adversary can impersonate the IMD or the reader to fool the other. Hence, communicating parties need to make sure whom they are talking to before disclosing important information.Authorization: once the confidentiality, integrity, and availability of IMDs are guaranteed, and the users (a human user or a device such as a reader) are authenticated, proper authorization to identify the privileges of these users’ proceeds. For instance, a doctor who may issue commands to the IMD should be distinct from a nurse who may only read information to monitor the patient.Non-repudiation: there are cases in which one party’s actions (knowingly or not) bring unwanted consequences. For instance, in an IMD-enabled health care system, there can be many participants in the process of diagnosing, monitoring, and treating patients. These professionals should not be able to repudiate the actions they took during the process so that, if anything terrible happened next, it is possible to know who did what.Session key agreement: communicating entities need to agree on a session key and use that key to encrypt the exchanged information. Session keys are symmetric keys that are primarily derived from another key (called a master key) to restrict ciphertexts and minimize the exposure of an attack. Furthermore, using session keys improves communication performance since these keys do not need to be stored and searched. Moreover, symmetric key encryption is faster.Perfect forward secrecy: satisfying this security requirement means the past sessions will not be compromised even if a master key is compromised. In the context of IMDs, if the long-term key is stolen, and if this is known, the key can be updated, and only minimal information would be disclosed while all past communications can be kept safe from future compromises.Emergency authentication: if we deal with patients with implanted devices, there can always be emergencies requiring human intervention. Emergency authentication is one of the paradox requirements since unauthorized users need to access the implants to override the authorization and authentication properties, which calls for a clear definition of an emergency.

Concerning privacy, there are at least five privacy requirements [12,37] that should be satisfied:Device-existence privacy: this privacy requirement challenges the protocol designers to conceal the device’s information of an IMD-enabled system and prohibit an adversary from learning its existence.Device-type privacy: in the cases where the presence of a device cannot be wholly concealed or its privacy cannot be maintained, the type of the device should stay anonymous. By doing so, it is possible to protect the patient from device-type specific attacks.Specific-device ID privacy: the unique ID (or serial number) of an IMD should not be disclosed to unauthorized users. Doing so protects the patients by prohibiting attackers from tracking down their locations.Measurement and log privacy: the information measured, collected, and analyzed in either IMD or the reader should be kept private. Keeping the privacy of logs enables the investigation and trace actions taken during the communication.Bearer privacy: these are often related to information such as patients’ names, record history, tests, IMD characteristics, etc., which should be kept private.

### 3.2. Security Issues and Proposed Solutions

Threats are only dangerous because of adversaries, malicious entities that usually have access to the communication media and are placed between the authorized entities to violate confidentiality (and privacy), integrity, and availability. These adversaries can be passive or active, internal or external, computationally restrained or unrestrained, and single individual vs. group [6].

In regard to IMD security, we can broadly classify adversaries based on their capabilities as passive eavesdroppers and active attackers [37,38,39]. The first class of adversaries can only eavesdrop on the radio communication between the legitimate entities to discover unencrypted messages. Sometimes, even if the messages are encrypted, passive adversaries may observe patterns to violate the privacy of communicating parties, such as learning the existence of IMD. 

On the other hand, active attackers can replay, modify, or delete messages in addition to possessing all of the capabilities of passive adversaries. These are the most dangerous types of adversaries that can bring life-threatening attacks to IMD-enabled systems. Adversaries in this category can execute replay attacks by forwarding exchanged messages later, changing critical settings of the implants by producing new commands, and exhausting the battery life of IMDs.

Different researchers have studied various security and privacy issues that challenge the normal operations of IMDs along with various proposed solutions that can be generally categorized as auditing-alone solutions, cryptographic solutions, and access control schemes [6]. The first category refers to solutions that solely depend on the access logs for the IMD. However, such techniques may not be suitable, as they cannot withstand active attacks if not used with other techniques such as access control mechanisms. The second measure utilizes cryptographic rudiments such as asymmetric-key cryptography, symmetric-key cryptography, and cryptographic hash functions. Three problems have been identified concerning the cryptographic solutions for IMDs [40]—the difficulty of implementation as most of the IMDs are already implanted in the human body, challenging to authenticate doctors during emergencies in which the patient is unconscious, and difficulty in maintaining the hardware and software of the implanted devices. The third solution refers to schemes that make use of access control help to protect IMDs from unauthorized access. The noticeable weakness in this solution is the difficulty of access during an emergency [6].

## 4. Formal Security Verification

Checking the safety of security protocols via a formal approach boosts users’ confidence, giving more convincing proof than its informal counterpart. When it comes to security protocols, such techniques may be divided into three categories: modal logic, model checkers, and theorem provers. This section will use one from the variants of modal logic (BAN logic) and another from model checking (AVISPA) to perform formal security verification for the authentication schemes proposed to safeguard IMDs. It is worth mentioning that the last two IMD authentication protocols (shown in Section 4.3.6 and Section 4.3.7) have also been analyzed, in [41,42], by the same authors. 

### 4.1. BAN Logic Based Formal Security Verification

BAN logic uses logic of beliefs to analyze authentication protocols by following its own rules. First, the messages exchanged between the participants of the protocol are idealized. Then, reasonable assumptions will be formulated, and the objectives that the protocol intends to meet are defined. Finally, a derivation step follows where the BAN logic rules are used together with the assumptions and the intermediate results to reach the goals. Figure 2 shows a typical procedure of carrying out formal analysis using BAN logic. The BAN logic symbols and rules are shown in Table 1 and Table 2, respectively.

### 4.2. AVISPA Based Formal Security Analysis

The previous section shows that BAN logic has been extensively used to verify authentication protocols by transforming them into a particular format and validating them through different logical rules. Unfortunately, BAN logic has limitations in accurately specifying a protocol in the idealization phase [21,43]. For that reason, most authentication protocols use automated formal security verification tools alongside BAN logic.

AVISPA provides a language called the high-level protocol specification language (HLPSL) [44] for describing security protocols and specifying their intended security properties, as well as a set of tools to validate them formally. An hlpsl2if translates the HLPSL specification into the Intermediate Format (IF). IF is a lower-level language that is read directly by the back-ends of the AVISPA Tool. The IF specification of a protocol is then input to the back-ends of the AVISPA Tool to analyze the stated security goals. Figure 3 shows this process.

The HLPSL specification is consists of basic roles, transitions, and composed roles used in three modules: role, session, and environment. Basic role refers to the specification of each of the modeled protocol participants and the initially known information as a parameter. These roles are then called to specify how the resulting participants interact by connecting various basic roles into a composed role. The transition part of an HLPSL specification encompasses a set of transitions between different roles. Each transition symbolizes the acceptance of a message and the sending of a response message.

### 4.3. Formal Security Analysis of IMD Authentication Protocols

#### 4.3.1. Khan et al.’s Protocol

The protocol proposed by Khan et al. [23] is a privacy-preserving key agreement protocol for WBANs. The protocol has four main participants: the system administrator (SA), the hub node (HN), the intermediary nodes (IN), and the normal nodes (N). HN is often considered a trusted high-end server that does not have computing resource constraints. The Ns are implanted sensors with computational limitations. The intermediary nodes have better processing, battery, and storage than Ns, and they are placed between HN and Ns to relay traffic. Furthermore, the protocol is executed in three main phases: initialization, registration, and authentication. Figure 4 shows the final phase of the protocol. Figure 5 presents the OFMC and CL-AtSe back-end results of the protocol.

BAN logic based Formal Security Analysis
●    **Idealization**(I1)$\mathrm{SN}\to \mathrm{HN}:{\;\langle \mathrm{id}{}^{\prime }}_{N},{\;r}_{N},{\;t}_{N},\mathrm{SN}\overset{x_{N}}{↔}\mathrm{HN}\rangle {}_{\mathrm{SN}\overset{{\mathrm{id}}_{N}}{↔}\mathrm{HN}}$(I2)$\mathrm{HN}\to \mathrm{SN}:{\;\langle r}_{N},f_{N},\mathrm{SN}\overset{x_{N}}{↔}\mathrm{HN},\;\mathrm{SN}\overset{\mathrm{ks}}{↔}\mathrm{HN}\rangle {}_{\mathrm{SN}\overset{{\mathrm{id}}_{N}}{↔}\mathrm{HN}}$●    **Assumption**(A1)$\mathrm{HN}\;\mathrm{believes}\;\mathrm{SN}\overset{{\mathrm{id}}_{N}}{↔}\mathrm{HN}$(A2)$\mathrm{HN}\;\mathrm{believes}\;\mathrm{fresh}\left(t_{N}\right)$(A3)${\mathrm{HN}\;\mathrm{believes}\;\mathrm{SN}\;\mathrm{controls}\;\mathrm{id}{}^{\prime }}_{N}$(A4)$\mathrm{HN}\;\mathrm{believes}\;\mathrm{SN}\;\mathrm{controls}\;\mathrm{SN}\overset{x_{N}}{↔}\mathrm{HN}$(A5)$\mathrm{SN}\;\mathrm{believes}\;\mathrm{SN}\overset{{\mathrm{id}}_{N}}{↔}\mathrm{HN}$(A6)$\mathrm{SN}\;\mathrm{believes}\;\mathrm{fresh}\left(f_{N}\right)$(A7)$\mathrm{SN}\;\mathrm{believes}\;\mathrm{HN}\;\mathrm{controls}\;\mathrm{SN}\overset{\mathrm{ks}}{↔}\mathrm{HN}$●    **Goals**(G1)${\mathrm{HN}\;\mathrm{believes}\;N\;\mathrm{belives}\;\mathrm{id}{}^{\prime }}_{N}$(G2)${\mathrm{HN}\;\mathrm{believes}\;\mathrm{id}{}^{\prime }}_{N}$(G3)$\mathrm{HN}\;\mathrm{believes}\;\mathrm{SN}\;\mathrm{belives}\;\mathrm{SN}\overset{x_{N}}{↔}\mathrm{HN}$(G4)$\mathrm{HN}\;\mathrm{belives}\;\mathrm{SN}\overset{x_{N}}{↔}\mathrm{HN}$(G5)$\mathrm{SN}\;\mathrm{believes}\;\mathrm{HN}\;\mathrm{belives}\;\mathrm{SN}\overset{x_{N}}{↔}\mathrm{HN}$(G6)$\mathrm{SN}\;\mathrm{believes}\;\mathrm{HN}\;\mathrm{belives}\;\mathrm{SN}\overset{\mathrm{ks}}{↔}\mathrm{HN}$(G7)$\mathrm{SN}\;\mathrm{belives}\;\mathrm{SN}\overset{\mathrm{ks}}{↔}\mathrm{HN}$
Figure 4Khan et al.’s protocol.
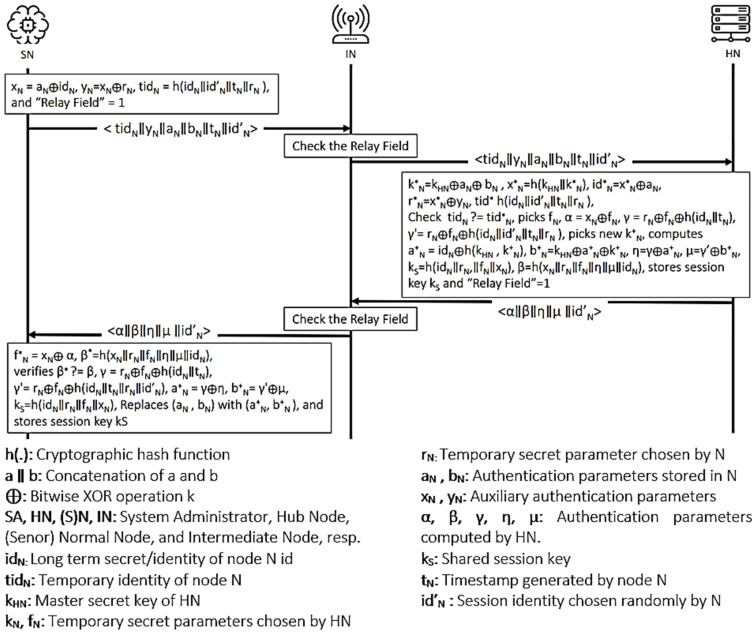

●    **Derivations**(D1)${\mathrm{HN}\;\mathrm{sees}\;\langle id{}^{\prime }}_{N},{\;r}_{N},{\;t}_{N},\mathrm{SN}\overset{x_{N}}{↔}\mathrm{HN}\rangle {}_{\mathrm{SN}\overset{{\mathrm{id}}_{N}}{↔}\mathrm{HN}}$(I1)(D2)$\mathrm{HN}\;\mathrm{believes}\;\mathrm{SN}\;\mathrm{said}\;\left[{id{}^{\prime }}_{N},{\;r}_{N},{\;t}_{N},\mathrm{SN}\overset{x_{N}}{↔}\mathrm{HN}\right]$By (D1),(A1),MM(D3)$\mathrm{HN}\;\mathrm{believes}\;\mathrm{SN}\;\mathrm{belives}\;\left[{id{}^{\prime }}_{N},{\;r}_{N},{\;t}_{N},\mathrm{SN}\overset{x_{N}}{↔}\mathrm{HN}\right]$By (D2),(A2),NV, FR(D4)${\mathrm{HN}\;\mathrm{believes}\;\mathrm{SN}\;\mathrm{belives}\;\mathrm{id}{}^{\prime }}_{N}$By (D3),BC(D5)${\mathrm{HN}\;\mathrm{believes}\;\mathrm{id}{}^{\prime }}_{N}$By (D4),(A3),JR. (D6)HN believes SN belives $\mathrm{SN}\overset{x_{N}}{↔}\mathrm{HN}$By (D3),BC(D7)$\mathrm{HN}\;\mathrm{belives}\;\mathrm{SN}\overset{x_{N}}{↔}\mathrm{HN}$By (D6), (A4), JR(D8)${\mathrm{SN}\;\mathrm{sees}\;\langle r}_{N},f_{N},\mathrm{SN}\overset{x_{N}}{↔}\mathrm{HN},\;\mathrm{SN}\overset{\mathrm{ks}}{↔}\mathrm{HN}\rangle {}_{\mathrm{SN}\overset{{\mathrm{id}}_{N}}{↔}\mathrm{HN}}$(I2)(D9)$\mathrm{SN}\;\mathrm{believes}\;\mathrm{HN}\;\mathrm{said}\;\left[{\;r}_{N},f_{N},\mathrm{SN}\overset{x_{N}}{↔}\mathrm{HN},\;\mathrm{SN}\overset{\mathrm{ks}}{↔}\mathrm{HN}\right]\;$By (D8),(A5),MM(D10)$\mathrm{SN}\;\mathrm{believes}\;\mathrm{HN}\;\mathrm{belives}\;\left[{\;r}_{N},f_{N},\mathrm{SN}\overset{x_{N}}{↔}\mathrm{HN},\;\mathrm{SN}\overset{\mathrm{ks}}{↔}\mathrm{HN}\right]$By (D9),(A6),NV, FR(D11)$\mathrm{SN}\;\mathrm{believes}\;\mathrm{HN}\;\mathrm{belives}\;\mathrm{SN}\overset{x_{N}}{↔}\mathrm{HN}$By (D10), BC(D12)$\mathrm{SN}\;\mathrm{believes}\;\mathrm{HN}\;\mathrm{believes}\;\mathrm{SN}\overset{\mathrm{ks}}{↔}\mathrm{HN}$By (D10), BC(D13)$\mathrm{SN}\;\mathrm{believes}\;\mathrm{SN}\overset{\mathrm{ks}}{↔}\mathrm{HN}$By (D12), (A7), JR
AVISPA based Formal Security Analysis Result
Figure 5AVISPA result of Khan et al.’s protocol.
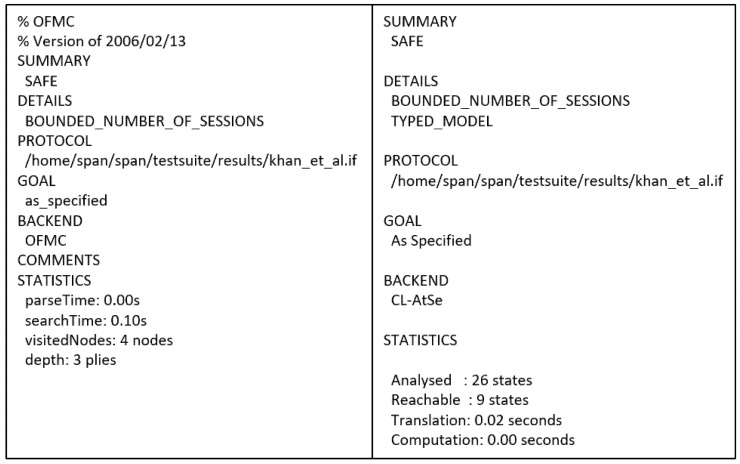



#### 4.3.2. Wu et al.’s Protocol

This protocol [24] is a proxy-based access control protocol that uses attribute-based encryption, particularly the ciphertext policy attribute-based encryption (CP-ABE). The protocol is executed by three participants—IMD, operator, and proxy. The IMDs have unique identifications ID_i_ and a master key K_i_^M^, which is only used for the initial pairing process with the proxy. All operators with the public parameters PK used in CP-ABE, unique identifications ID_o_, a public and private key pair (PU_OP_ and PR_OP_, respectively), and a certificate Cert must first be registered at a Central Health Authority (CHA). The CHA will then generate the secret key SK. The operator uses a programmer to communicate with the IMD and proxy after it obtains the required information by manual inputting or reading in from a smart card. With the identification of ID_p_ and connection with an IMD programmer through an audio cable, the proxy device performs the access control for the IMD. Figure 6 shows the flow of messages in the protocol. Figure 7 illustrates the OFMC and CL-AtSe back-end results of the protocol.

BAN logic-based formal security analysis.
Figure 6Wu et al.’s protocol.
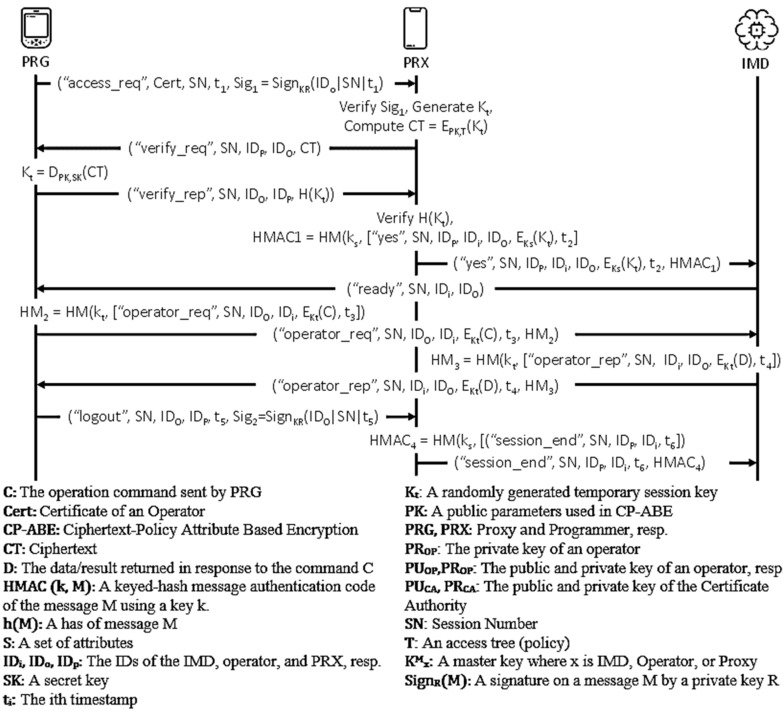

●    **Idealization**(I1)$\mathrm{PRG}\to \mathrm{PRX}:\langle {\mathrm{ID}}_{O},\mathrm{SN},{\;t}_{1}\rangle {}_{{\mathrm{PU}}^{-1}{}_{\mathrm{OP}}}$(I2)$\mathrm{PRX}\to \mathrm{PRG}:\langle \mathrm{PRG}\overset{K_{t}}{↔}\mathrm{IMD}\rangle {}_{\mathrm{PKT}}$(I3)$\mathrm{PRX}\to \mathrm{IMD}:\langle \mathrm{PRG}\overset{K_{t}}{↔}\mathrm{IMD},\rangle {}_{\mathrm{PRX}\overset{K_{S}}{↔}\mathrm{IMD}}$, $\langle \mathrm{SN},{\;\mathrm{ID}}_{P},{\mathrm{ID}}_{I},{\;\mathrm{ID}}_{O},{\;t}_{2}\rangle {}_{\mathrm{PRX}\overset{K_{S}}{↔}\mathrm{IMD}}$(I4)$\mathrm{PRG}\to \mathrm{IMD}:\langle \mathrm{PRG}\overset{K_{t}}{↔}\mathrm{IMD},\;C\rangle {}_{\mathrm{PRG}\overset{K_{t}}{↔}\mathrm{IMD}}$, $\langle \;\mathrm{PRG}\overset{K_{t}}{↔}\mathrm{IMD},\;\mathrm{SN},{\;\mathrm{ID}}_{O},{\mathrm{ID}}_{I},{\;t}_{3}\rangle {}_{\mathrm{PRG}\overset{K_{t}}{↔}\mathrm{IMD}}$(I5)$\mathrm{IMD}\to \mathrm{PRG}:\langle \mathrm{PRG}\overset{K_{t}}{↔}\mathrm{IMD},\;D\rangle {}_{\mathrm{PRG}\overset{K_{t}}{↔}\mathrm{IMD}}$, $\langle \;\mathrm{PRG}\overset{K_{t}}{↔}\mathrm{IMD},\;\mathrm{SN},{\;\mathrm{ID}}_{I},{\mathrm{ID}}_{O},{\;t}_{4}\rangle {}_{\mathrm{PRG}\overset{K_{t}}{↔}\mathrm{IMD}}$●    **Assumption**(A1)$\mathrm{PRX}\;\mathrm{believes}\;\overset{{\mathrm{PU}}_{\mathrm{OP}}}{\to }\mathrm{PRG}$
(A2)$\mathrm{PRX}\;\mathrm{believes}\;\mathrm{fresh}\left(t_{1}\right)$
(A3)${\mathrm{PRX}\;\mathrm{believes}\;\mathrm{PRG}\;\mathrm{Controls}\;\mathrm{ID}}_{O}$
(A4)$\mathrm{PRX}\;\mathrm{believes}\;\overset{{\mathrm{PKT}}^{-1}}{\to }\mathrm{PRG}$
(A5)$\mathrm{PRG}\;\mathrm{believes}\;\mathrm{fresh}\left(K_{t}\right)$
(A6)$\mathrm{PRG}\;\mathrm{believes}\;\mathrm{PRX}\;\mathrm{Controls}\;\mathrm{PRX}\overset{K_{t}}{↔}\mathrm{PRG}$
(A7)$\mathrm{IMD}\;\mathrm{believes}\;\mathrm{PRX}\overset{K_{s}}{↔}\mathrm{IMD}$
(A8)$\mathrm{IMD}\;\mathrm{believes}\;\mathrm{fresh}\;\left(K_{t}\right)$
(A9)$\mathrm{IMD}\;\mathrm{believes}\;\mathrm{fresh}\left(t_{2}\right)$
(A10)$\mathrm{IMD}\;\mathrm{believes}\;\mathrm{PRX}\;\mathrm{Controls}\;\mathrm{PRX}\overset{K_{t}}{↔}\mathrm{IMD}$
(A11)${\mathrm{IMD}\;\mathrm{believes}\;\mathrm{PRX}\;\mathrm{Controls}\;\mathrm{ID}}_{P}$
(A12)$\mathrm{IMD}\;\mathrm{believes}\;\mathrm{fresh}\left(t_{3}\right)$
(A13)${\mathrm{IMD}\;\mathrm{believes}\;\mathrm{PRG}\;\mathrm{Controls}\;\mathrm{ID}}_{O}$
(A14)$\mathrm{IMD}\;\mathrm{believes}\;\mathrm{fresh}\left(t_{4}\right)$
(A15)${\mathrm{PRG}\;\mathrm{believes}\;\mathrm{IMD}\;\mathrm{Controls}\;\mathrm{ID}}_{I}$
●    **Goals**(G1)${\mathrm{PRX}\;\mathrm{believes}\;\mathrm{ID}}_{O}$
(G2)$\mathrm{PRG}\;\mathrm{believes}\;\mathrm{PRX}\;\mathrm{believes}\;\mathrm{PRG}\overset{K_{t}}{↔}\mathrm{IMD}$
(G3)$\mathrm{PRG}\;\mathrm{believes}\;\mathrm{PRG}\overset{K_{t}}{↔}\mathrm{IMD}$
(G4)$\mathrm{IMD}\;\mathrm{believes}\;\mathrm{PRX}\;\mathrm{believes}\;\mathrm{PRG}\overset{K_{t}}{↔}\mathrm{IMD}$
(G5)$\mathrm{IMD}\;\mathrm{believes}\;\mathrm{PRG}\overset{K_{t}}{↔}\mathrm{IMD}$
(G6)${\mathrm{IMD}\;\mathrm{believes}\;\mathrm{ID}}_{P}$
(G7)${\mathrm{IMD}\;\mathrm{believes}\;\mathrm{ID}}_{O}$
(G8)$\mathrm{IMD}\;\mathrm{believes}\;\mathrm{PRG}\;\mathrm{believes}\;\mathrm{PRG}\overset{K_{t}}{↔}\mathrm{IMD}$
(G9)$\mathrm{PRG}\;\mathrm{believes}\;\mathrm{IMD}\;\mathrm{believes}\;\mathrm{PRG}\overset{K_{t}}{↔}\mathrm{IMD}$
(G10)${\mathrm{PRG}\;\mathrm{believes}\;\mathrm{ID}}_{I}$
●    **Derivations**(D1)${\mathrm{PRX}\;\mathrm{sees}\;\langle \mathrm{ID}}_{O},\mathrm{SN},{\;t}_{1}\rangle {}_{{\mathrm{PU}}^{-1}{}_{\mathrm{OP}}}$(I1)(D2)$\mathrm{PRX}\;\mathrm{believes}\;\mathrm{PRG}\;\mathrm{said}\;\left[{\mathrm{ID}}_{O},\;\mathrm{SN},{\;t}_{1}\right]$By (D1),(A1),MM(D3)$\mathrm{PRX}\;\mathrm{believes}\;\mathrm{PRG}\;\mathrm{believes}\;\left[{\mathrm{ID}}_{O},\;\mathrm{SN},{\;t}_{1}\right]\;$By (D2),(A2),NV, FR(D4)${\mathrm{PRX}\;\mathrm{believes}\;\mathrm{PRG}\;\mathrm{believes}\;\mathrm{ID}}_{O}$By (D3),BC(D5)${\mathrm{PRX}\;\mathrm{believes}\;\mathrm{ID}}_{O}$By (D4),(A3),JR(D6)$\mathrm{PRG}\;\mathrm{sees}\;\langle \mathrm{PRG}\;\overset{K_{t}}{↔}\mathrm{IMD}\rangle {}_{\mathrm{PKT}}$(I2)(D7)$\mathrm{PRG}\;\mathrm{believes}\;\mathrm{PRX}\;\mathrm{said}\;\mathrm{PRG}\overset{K_{t}}{↔}\mathrm{IMD}$By (D6), (A4), JR(D8)$\mathrm{PRG}\;\mathrm{believes}\;\mathrm{PRX}\;\mathrm{believes}\;\mathrm{PRG}\overset{K_{t}}{↔}\mathrm{IMD}$By (D7), (A5), NV, FR(D9)$\mathrm{PRG}\;\mathrm{believes}\;\mathrm{PRG}\overset{K_{t}}{↔}\mathrm{IMD}$By (D8),(A6),JR(D10)$\mathrm{IMD}\;\mathrm{sees}\;\langle \mathrm{PRG}\overset{K_{t}}{↔}\mathrm{IMD}\rangle {}_{\mathrm{PRX}\overset{K_{S}}{↔}\mathrm{IMD}\;},\;\langle \mathrm{SN},{\;\mathrm{ID}}_{P},{\mathrm{ID}}_{I},{\;\mathrm{ID}}_{O},{\;t}_{2}\rangle {}_{\mathrm{PRX}\overset{K_{S}}{↔}\mathrm{IMD}}\;$(I3)(D11)$\mathrm{IMD}\;\mathrm{believes}\;\mathrm{PRX}\;\mathrm{said}\;\mathrm{PRG}\overset{K_{t}}{↔}\mathrm{IMD}$By (D10), DR, (A7),MM(D12)$\mathrm{IMD}\;\mathrm{believes}\;\mathrm{PRX}\;\mathrm{said}\;\left[\mathrm{SN},{\;\mathrm{ID}}_{P},{\mathrm{ID}}_{I},{\;\mathrm{ID}}_{O},{\;t}_{2}\right]\;$By (D10), DR, (A7),MM(D13)$\mathrm{IMD}\;\mathrm{believes}\;\mathrm{PRX}\;\mathrm{believes}\;\mathrm{PRG}\overset{K_{t}}{↔}\mathrm{IMD}\;$By (D11), (A8), NV, FR(D14)$\mathrm{IMD}\;\mathrm{believes}\;\mathrm{PRX}\;\mathrm{believes}\;\left[\mathrm{SN},{\;\mathrm{ID}}_{P},{\mathrm{ID}}_{I},{\;\mathrm{ID}}_{O},{\;t}_{2}\right]\;$By (D12), (A9), NV, FR(D15)${\mathrm{IMD}\;\mathrm{believes}\;\mathrm{PRX}\;\mathrm{believes}\;\mathrm{ID}}_{P}\;$By (D14), BC(D16)$\mathrm{IMD}\;\mathrm{believes}\;\mathrm{PRG}\overset{K_{t}}{↔}\mathrm{IMD}\;$By (D13), (A10), JR(D17)${\mathrm{IMD}\;\mathrm{believes}\;\mathrm{ID}}_{P}\;$By (D15), (A11), JR(D18)$\mathrm{IMD}\;\mathrm{sees}\;\langle \mathrm{PRG}\overset{K_{t}}{↔}\mathrm{IMD},\;\mathrm{SN},{\;\mathrm{ID}}_{O},{\mathrm{ID}}_{I},{\;t}_{3}\rangle {}_{\mathrm{PRG}\overset{K_{t}}{↔}\mathrm{IMD}}\;$By (I4), DR(D19)$\mathrm{IMD}\;\mathrm{believes}\;\mathrm{PRG}\;\mathrm{said}\;\left[\mathrm{PRG}\overset{K_{t}}{↔}\mathrm{IMD},\;\mathrm{SN},{\;\mathrm{ID}}_{O},{\mathrm{ID}}_{I},{\;t}_{3}\right]\;$By (D18), (D16), MM(D20)$\mathrm{IMD}\;\mathrm{believes}\;\mathrm{PRG}\;\mathrm{believes}\;\left[\mathrm{PRG}\overset{K_{t}}{↔}\mathrm{IMD},\;\mathrm{SN},{\;\mathrm{ID}}_{O},{\mathrm{ID}}_{I},{\;t}_{3}\right]\;$By (D19), (A12), NV, FR(D21)${\;\mathrm{IMD}\;\mathrm{believes}\;\mathrm{PRG}\;\mathrm{believes}\;\mathrm{ID}}_{O}$By (D20), BC(D22)${\mathrm{IMD}\;\mathrm{believes}\;\mathrm{ID}}_{O}$By (D21), (A13),JR(D23)$\mathrm{IMD}\;\mathrm{believes}\;\mathrm{PRG}\;\mathrm{believes}\;\mathrm{PRG}\overset{K_{t}}{↔}\mathrm{IMD}$By (D20), BC(D24)$\mathrm{PRG}\;\mathrm{sees}\;\langle \mathrm{PRG}\overset{K_{t}}{↔}\mathrm{IMD},\;\mathrm{SN},{\;\mathrm{ID}}_{I},{\mathrm{ID}}_{O},{\;t}_{4}\rangle {}_{\mathrm{PRG}\overset{K_{t}}{↔}\mathrm{IMD}}$By (I5), DR(D25)$\mathrm{PRG}\;\mathrm{believes}\;\mathrm{IMD}\;\mathrm{said}\;\left[\mathrm{PRG}\overset{K_{t}}{↔}\mathrm{IMD},\;\mathrm{SN},{\;\mathrm{ID}}_{I},{\mathrm{ID}}_{O},{\;t}_{4}\right]\;$By (D24), (D9), MM(D26)$\mathrm{PRG}\;\mathrm{believes}\;\mathrm{IMD}\;\mathrm{believes}\;\left[\mathrm{PRG}\overset{K_{t}}{↔}\mathrm{IMD},\;\mathrm{SN},{\;\mathrm{ID}}_{I},{\mathrm{ID}}_{O},{\;t}_{4}\right]\;$By (D25), (A14), NV, FR(D27)$\mathrm{PRG}\;\mathrm{believes}\;\mathrm{IMD}\;\mathrm{believes}\;\mathrm{PRG}\overset{K_{t}}{↔}\mathrm{IMD}$By (D26), BC(D28)${\mathrm{PRG}\;\mathrm{believes}\;\mathrm{IMD}\;\mathrm{believes}\;\mathrm{ID}}_{I}\;$By (D26), BC(D29)${\mathrm{PRG}\;\mathrm{believes}\;\mathrm{ID}}_{I}$
By (D28),(I15), JR
AVISPA-based formal security analysis result.
Figure 7AVISPA result of Wu et al.’s protocol.
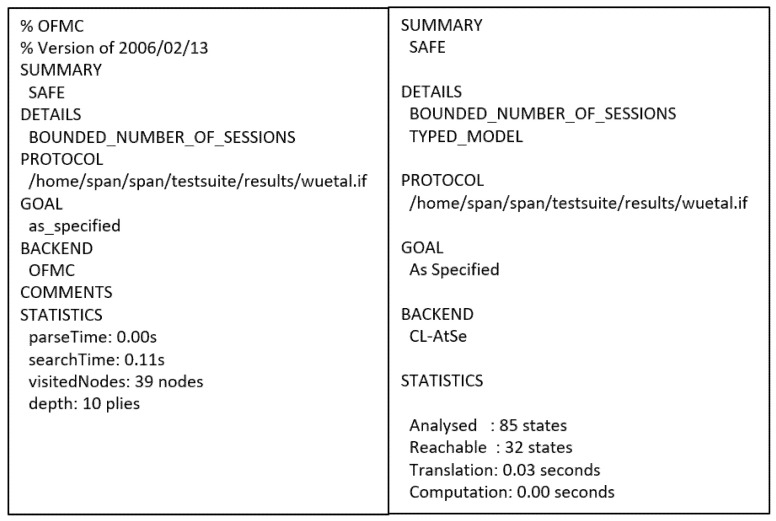



#### 4.3.3. Chi et al.’s Protocol

This protocol [25] uses a compressing-based encryption mechanism and public key infrastructure, and other cryptographic protocols, such as RSA, AES, and HMAC. The protocol comprises three participants—IMD, smartphone, and programmer. The IMD communicates with the patient’s smartphone via Bluetooth, and it interacts with the doctor’s programmer through the wireless medium. The smartphone refers to both the patient and doctor smartphones, in which the patient’s smartphone links with the IMD utilizing Bluetooth and connects with a programmer wirelessly. The protocol involves four stages—initialization, pairing, authentication, and authorization, as shown in Figure 8 and Figure 9 presents the OFMC and CL-AtSe back-end results of the protocol.

BAN logic-based formal security analysis.
●    **Idealization**(I1)$S\to I:\langle R_{1},{\mathrm{SN}}_{O},{\mathrm{ID}}_{S},{\mathrm{TS}}_{1},S\overset{K_{i}}{↔}I\rangle {}_{S\overset{K_{i}}{↔}I}$(I2)$D\to S:\langle R_{3},\mathrm{SN},{\mathrm{ID}}_{D},{\mathrm{TS}}_{2}\rangle {}_{{\mathrm{PK}}^{-1}}$(I3)$\;S\to D:\langle \mathrm{nonce}\rangle {}_{\mathrm{PK}}$(I4)$D\to S:\langle R_{4},\mathrm{SN},{\mathrm{ID}}_{D},{\mathrm{ID}}_{S},{\mathrm{TS}}_{3},D\overset{K_{p}}{↔}S\rangle {}_{D\overset{K_{p}}{↔}S}$(I5)$S\to I:\langle {\;R}_{6},\mathrm{SN},{\mathrm{ID}}_{D},C_{1},{\mathrm{TS}}_{5}{}_{S\overset{K_{i}}{↔}I},I\overset{K_{d}}{↔}D,I\overset{K_{r}}{↔}D\rangle {}_{S\overset{K_{i}}{↔}I}$(I6)$S\to D:\langle I\overset{K_{d}}{↔}D\rangle {}_{D\overset{K_{p}}{↔}S}$(I7)$I\to D:\langle \mathrm{SN},C_{2},I\overset{K_{r}}{↔}D,\rangle {}_{I\overset{K_{r}}{↔}D},\langle \mathrm{data},I\overset{K_{d}}{↔}D\rangle {}_{I\overset{K_{d}}{↔}D}$(I8)$I\to S:\langle \mathrm{CMD},{\mathrm{ID}}_{I},{\mathrm{TS}}_{7}\rangle {}_{S\overset{K_{i}}{↔}I}$●    **Assumption**(A1)$I\;\mathrm{believes}\;S\overset{K}{↔}I$
(A2)${I\;\mathrm{believes}\;\mathrm{ID}}_{I}$
(A3)${I\;\mathrm{believes}\;\mathrm{ID}}_{S}$
(A4)$I\;\mathrm{believes}\;\mathrm{fresh}\left({\mathrm{TS}}_{1}\right)$
(A5)$S\;\mathrm{believes}\;\overset{\mathrm{PK}}{\to }D$
(A6)$S\;\mathrm{believes}\;\mathrm{fresh}\left({\mathrm{TS}}_{2}\right)$
(A7)S believes D Controls SN
(A8)$D\;\mathrm{believes}\;\overset{\mathrm{PK}}{\to }D$
(A9)D believes fresh(nonce)
(A10)D believes S Controls nonce
(A11)D believes RM
(A12)D believes SN
(A13)S believes RM
(A14)S believes nonce
(A15)$S\;\mathrm{believes}\;\mathrm{fresh}\left({\mathrm{TS}}_{3}\right)$
(A16)$I\;\mathrm{believes}\;\mathrm{fresh}\left(I\overset{K_{d}}{↔}D\right)$
(A17)$I\;\mathrm{believes}\;S\;\mathrm{Controls}\;I\overset{K_{d}}{↔}D$
(A18)$I\;\mathrm{believes}\;S\;\mathrm{Controls}\;I\overset{K_{r}}{↔}D$
(A19)$D\;\mathrm{believes}\;\mathrm{fresh}\left(I\overset{K_{d}}{↔}D\right)$
(A20)$D\;\mathrm{believes}\;S\;\mathrm{Controls}\;I\overset{K_{d}}{↔}D$
(A21)$D\;\mathrm{believes}\;\mathrm{fresh}\left(I\overset{K_{r}}{↔}D\right)$
(A22)$S\;\mathrm{believes}\;S\overset{K_{i}}{↔}I$
(A23)$S\;\mathrm{believes}\;\mathrm{fresh}\left({\mathrm{TS}}_{7}\right)$
●    **Goals**(G1)$I\;\mathrm{believes}\;S\;\mathrm{belives}\;S\overset{K_{i}}{↔}I$
(G2)$I\;\mathrm{belives}\;S\overset{K_{i}}{↔}I$
(G3)$S\;\mathrm{believes}\;I\;\mathrm{belives}\;S\overset{K_{i}}{↔}I$
(G4)$D\;\mathrm{believes}\;S\;\mathrm{belives}\;D\overset{K_{p}}{↔}S$
(G5)$D\;\mathrm{belives}\;D\overset{K_{p}}{↔}S$
(G6)$S\;\mathrm{believes}\;D\;\mathrm{belives}\;D\overset{K_{p}}{↔}S$
(G7)$S\;\mathrm{belives}\;D\overset{K_{p}}{↔}S$
(G8)$I\;\mathrm{believes}\;S\;\mathrm{belives}\;D\overset{K_{d}}{↔}S$
(G9)$I\;\mathrm{believes}\;S\;\mathrm{belives}\;D\overset{K_{r}}{↔}S$
(G10)$I\;\mathrm{belives}\;D\overset{K_{d}}{↔}S$
(G11)$I\;\mathrm{belives}\;D\overset{K_{r}}{↔}S$
(G12)$\;D\;\mathrm{believes}\;S\;\mathrm{belives}\;D\overset{K_{d}}{↔}S$
(G13)$D\;\mathrm{belives}\;D\overset{K_{d}}{↔}S$
(G14)$D\;\mathrm{belives}\;D\overset{K_{r}}{↔}S$
(G15)$I\;\mathrm{believes}\;D\;\mathrm{belives}\;D\overset{K_{d}}{↔}S$
(G16)$I\;\mathrm{believes}\;D\;\mathrm{belives}\;D\overset{K_{r}}{↔}S$
(G17)$D\;\mathrm{believes}\;I\;\mathrm{belives}\;D\overset{K_{d}}{↔}S$
●    **Derivations**(D1)${I\;\mathrm{sees}\;\langle R}_{1},{\;\mathrm{SN}}_{O},{\mathrm{ID}}_{S},{\mathrm{TS}}_{1},S\overset{K_{i}}{↔}I\rangle {}_{S\overset{K_{i}}{↔}I}\;$(I1)(D2)$I\;\mathrm{believes}\;S\overset{K_{i}}{↔}I\;$(A1),(A2),(A3), BC(D3)$I\;\mathrm{believes}\;S\;\mathrm{said}\;\left[R_{1},{\;\mathrm{SN}}_{O},{\mathrm{ID}}_{S},{\mathrm{TS}}_{1},S\overset{K_{i}}{↔}I\right]\;$By (D1),(D2),MM(D4)$I\;\mathrm{believes}\;S\;\mathrm{believes}\;\left[R_{1},{\;\mathrm{SN}}_{O},{\mathrm{ID}}_{S},{\mathrm{TS}}_{1},S\overset{K_{i}}{↔}I\right]\;$By (D2),(A4),NV, FR(D5)$I\;\mathrm{believes}\;S\;\mathrm{believes}\;S\overset{K_{i}}{↔}I$By (D4),BC
(D6)${S\;\mathrm{sees}\;\langle R}_{3},\;\mathrm{SN},{\mathrm{ID}}_{D},{\mathrm{TS}}_{2}\rangle {}_{{\mathrm{PK}}^{-1}}\;$(I2)(D7)$S\;\mathrm{believes}\;D\;\mathrm{said}\;\left[R_{3},\;\mathrm{SN},{\mathrm{ID}}_{D},{\mathrm{TS}}_{2}\right]\;$By (D6), (A5), MM(D8)$S\;\mathrm{believes}\;D\;\mathrm{believes}\;\left[R_{3},\;\mathrm{SN},{\mathrm{ID}}_{D},{\mathrm{TS}}_{2}\right]\;$By (D7), (A6), NV, FR
(D9)S believes D believes SN By (D8), BC(D10)S believes SN By (D9), (A7), JR(D11)$D\;\mathrm{sees}\;\langle \mathrm{nonce}\rangle {}_{\mathrm{PK}}\;$(I3)(D12)D believes S said [nonce] By (D11), (A8), MM(D13)D believes S believes nonce By (D12),(A9),NV, FR(D14)D believes nonce By (D13),(A10),JR(D15)$D\;\mathrm{belives}\;D\overset{K_{p}}{↔}S\;$By (D14),(A11),(A12), BC(D16)$S\;\mathrm{belives}\;D\overset{K_{p}}{↔}S\;$By (D10),(A13),(A14),BC(D17)${S\;\mathrm{sees}\;\langle R}_{4},\;\mathrm{SN},{\mathrm{ID}}_{D},{\;\mathrm{ID}}_{S},{\mathrm{TS}}_{3},D\overset{K_{p}}{↔}S\rangle {}_{D\overset{K_{p}}{↔}S}$(I4)(D18)$S\;\mathrm{believes}\;D\;\mathrm{said}\;\left[R_{4},\;\mathrm{SN},{\mathrm{ID}}_{D},{\;\mathrm{ID}}_{S},{\mathrm{TS}}_{3},D\overset{K_{p}}{↔}S\right]$By (D17), (D16), MM
(D19)$S\;\mathrm{believes}\;D\;\mathrm{belives}\;\left[R_{4},\;\mathrm{SN},{\mathrm{ID}}_{D},{\;\mathrm{ID}}_{S},{\mathrm{TS}}_{3},D\overset{K_{p}}{↔}S\right]\;$By (D18), (A15), NV, FR(D20)$S\;\mathrm{believes}\;D\;\mathrm{believes}\;D\overset{K_{p}}{↔}S\;$By (D19), BC(D21)${I\;\mathrm{sees}\;\langle R}_{6},\;\mathrm{SN},{\mathrm{ID}}_{D},C_{1},{\;\mathrm{TS}}_{5}\rangle {}_{S\overset{K_{i}}{↔}I},\langle I\overset{K_{d}}{↔}D,I\overset{K_{r}}{↔}D\rangle {}_{S\overset{K_{i}}{↔}I}$(I5)(D22)$I\;\mathrm{sees}\;\langle I\overset{K_{d}}{↔}D,I\overset{K_{r}}{↔}D\rangle {}_{S\overset{K_{i}}{↔}I}\;$By (D21),DR(D23)$I\;\mathrm{believes}\;S\;\mathrm{said}\;\left[I\overset{K_{d}}{↔}D,I\overset{K_{r}}{↔}D\right]\;$By (D22), (D2), MM(D24)$I\;\mathrm{believes}\;S\;\mathrm{belives}\;\left[I\overset{K_{d}}{↔}D,I\overset{K_{r}}{↔}D\right]\;$By (D23), (A16), NV, FR(D25)$I\;\mathrm{believes}\;S\;\mathrm{belives}\;I\overset{K_{d}}{↔}D\;$By (D24), BC(D26)$I\;\mathrm{believes}\;S\;\mathrm{belives}\;I\overset{K_{r}}{↔}D\;$By (D24), BC(D27)$I\;\mathrm{believes}\;I\overset{K_{d}}{↔}D\;$By (D25), (A17), JR(D28)$I\;\mathrm{believes}\;I\overset{K_{r}}{↔}D\;$By (D26), (A18), JR(D29)$D\;\mathrm{sees}\;\langle I\overset{K_{d}}{↔}D\rangle {}_{D\overset{K_{p}}{↔}S}\;$(I6)(D30)$D\;\mathrm{believes}\;S\;\mathrm{said}\;\left[I\overset{K_{d}}{↔}D\right]\;$By (D30), (A15), MM(D31)$D\;\mathrm{believes}\;S\;\mathrm{belives}\;\left[I\overset{K_{d}}{↔}D\right]\;$By (D30), (A19), NV, FR(D32)$D\;\mathrm{believes}\;S\;\mathrm{belives}\;I\overset{K_{d}}{↔}D\;$By (D31), BC(D33)$D\;\mathrm{believes}\;I\overset{K_{d}}{↔}D\;$By (D32), (A20), JR(D34)$D\;\mathrm{sees}\;\langle \mathrm{SN},{\;C}_{2},\;I\overset{K_{r}}{↔}D\rangle {}_{I\overset{K_{r}}{↔}D},\langle \mathrm{data},\;I\overset{K_{d}}{↔}D\rangle {}_{I\overset{K_{d}}{↔}D}\;$(I7)(D35)$D\;\mathrm{sees}\;\langle \mathrm{data},\;I\overset{K_{d}}{↔}D\rangle {}_{I\overset{K_{d}}{↔}D}\;$By (D34), DR(D36)$D\;\mathrm{believes}\;I\;\mathrm{said}\;\left[\mathrm{data},\;I\overset{K_{d}}{↔}D\right]\;$By (D35), (D33), MM(D37)$D\;\mathrm{believes}\;I\;\mathrm{belives}\;\left[\mathrm{data},\;I\overset{K_{d}}{↔}D\right]\;$By (D30), (A19), NV, FR(D38)$D\;\mathrm{believes}\;I\;\mathrm{belives}\;I\overset{K_{d}}{↔}D\;$By (D37), BC(D39)$D\;\mathrm{sees}\;\langle \mathrm{SN},{\;C}_{2},\;I\overset{K_{r}}{↔}D\rangle {}_{I\overset{K_{r}}{↔}D}\;$By (D34), DR(D40)$D\;\mathrm{believes}\;I\overset{K_{r}}{↔}D\;$By (D33), (A12), BC(D41)$D\;\mathrm{believes}\;I\;\mathrm{said}\;\left[\mathrm{SN},{\;C}_{2},\;I\overset{K_{r}}{↔}D\right]\;$By (D39), (D40), MM(D42)$D\;\mathrm{believes}\;I\;\mathrm{believes}\;\left[\mathrm{SN},{\;C}_{2},\;I\overset{K_{r}}{↔}D\right]\;$By (D41), (A21), NV, FR(D43)$D\;\mathrm{believes}\;I\;\mathrm{believes}\;I\overset{K_{r}}{↔}D\;$By (D42), BC(D44)$S\;\mathrm{sees}\;\langle \mathrm{CMD},{\;\mathrm{ID}}_{I},{\mathrm{TS}}_{7},\;S\overset{K_{i}}{↔}I\rangle {}_{S\overset{K_{i}}{↔}I}\;$(I8)(D45)$S\;\mathrm{believes}\;I\;\mathrm{said}\;\left[\mathrm{CMD},{\;\mathrm{ID}}_{I},{\mathrm{TS}}_{7},\;S\overset{K_{i}}{↔}I\right]\;$By (D44), (A22), MM(D46)$S\;\mathrm{believes}\;I\;\mathrm{belives}\;\left[\mathrm{CMD},{\;\mathrm{ID}}_{I},{\mathrm{TS}}_{7},\;S\overset{K_{i}}{↔}I\right]\;$By (D45), (A23), NV, FR(D47)$S\;\mathrm{believes}\;I\;\mathrm{belives}\;S\overset{K_{i}}{↔}I\;$By (D46), BCAVISPA-based formal security analysis result.
Figure 9AVISPA result of Chi et al.’s protocol.
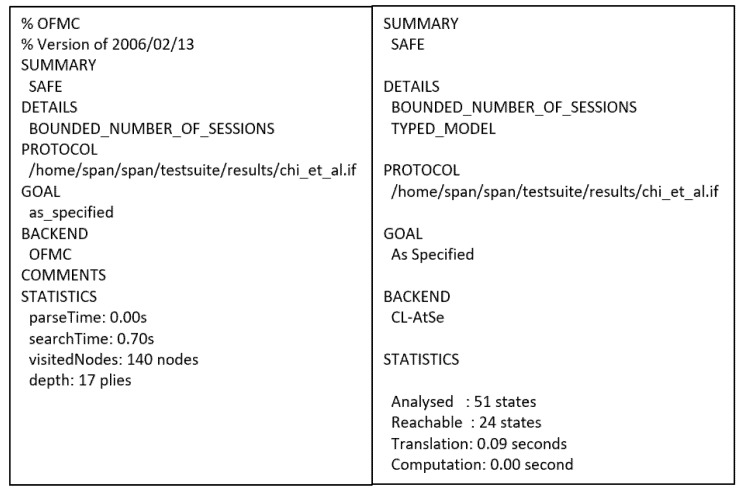


#### 4.3.4. Parvez et al.’s Protocol

The proposed authentication scheme [26] extended the protocol in [45] that comprises of sensors, which are resource-constrained devices that are implanted in (or wearable on) human body; mobile devices, which are small handheld devices to collect the data sent by the sensors; gateway, which is a trusted server that is used to register sensors, mobile devices and medical experts, and generates different keys for secure communication; and medical experts refers to medical professionals, such as doctors or nurses who analyze and take action with the collected information. The proposed protocol is executed in two phases—registration and authentication—as shown in Figure 10 and Figure 11 illustrates the OFMC and CL-AtSe back-end results of the protocol.

BAN logic-based formal security analysis.
●    **Idealization**(I1)$\mathrm{ME}\to \mathrm{GW}:\langle {\;M}_{\mathrm{id}},\mathrm{nonce},{\;U}_{i},{\mathrm{SN}}_{j},t_{1}\rangle {}_{\mathrm{GW}\overset{K_{l}}{↔}\mathrm{ME}},\langle M_{\mathrm{id}},{\;\mathrm{ID}}_{\mathrm{gw}}\rangle {}_{\mathrm{GW}\overset{K_{j}}{↔}\mathrm{ME}}$(I2)$\mathrm{GW}\to \mathrm{MD}:\langle {\;M}_{\mathrm{id}},{\;U}_{i},{\mathrm{SN}}_{j},\mathrm{nonce},t_{3}\rangle {}_{\mathrm{GW}\overset{K_{\mathrm{GW}-U}}{↔}\mathrm{MD}}$(I3)$\mathrm{MD}\to \mathrm{IMD}:\langle {\;M}_{\mathrm{id}},{\;U}_{i},{\mathrm{SN}}_{j},\mathrm{nonce},t_{5}\rangle {}_{\mathrm{MD}\overset{K_{U-{\mathrm{SN}}_{j}}}{↔}\mathrm{IMD}}$(I4)$\mathrm{IMD}\to \mathrm{ME}:\langle {\;\mathrm{SN}}_{j},M_{\mathrm{id}},\;\mathrm{IMD}\overset{K_{\mathrm{ssk}}}{↔}\mathrm{ME},t_{7}\rangle {}_{\mathrm{IMD}\overset{K_{\mathrm{ssk}}}{↔}\mathrm{ME}}$
Figure 10Parvez et al.’s protocol. (**a**) Registration procedure. (**b**) Authentication Procedure.
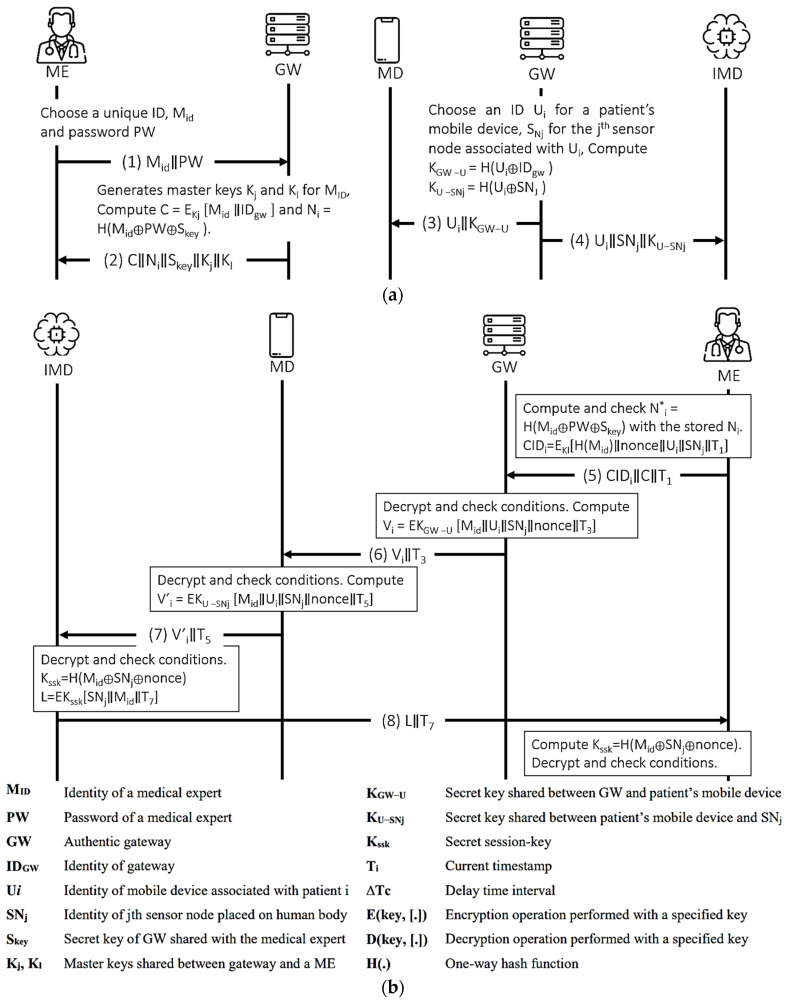

●    **Assumption**(A1)$\mathrm{GW}\;\mathrm{believes}\;\mathrm{GW}\overset{K_{l}}{↔}\mathrm{ME}$
(A2)$\mathrm{GW}\;\mathrm{believes}\;\mathrm{fresh}\left(t_{1}\right)$
(A3)GW believes ME Controls nonce
(A4)$\mathrm{MD}\;\mathrm{believes}\;\mathrm{GW}\overset{K_{\mathrm{GW}-U}}{↔}\mathrm{MD}$
(A5)$\mathrm{MD}\;\mathrm{believes}\;\mathrm{fresh}\left(t_{3}\right)$
(A6)${\mathrm{MD}\;\mathrm{believes}\;\mathrm{GW}\;\mathrm{Controls}\;\mathrm{SN}}_{j}$
(A7)${\mathrm{MD}\;\mathrm{believes}\;\mathrm{GW}\;\mathrm{Controls}\;M}_{\mathrm{id}}$
(A8)$\mathrm{IMD}\;\mathrm{believes}\;\mathrm{MD}\overset{K_{U-{\mathrm{SN}}_{j}}}{↔}\mathrm{IMD}$
(A9)$\mathrm{IMD}\;\mathrm{believes}\;\mathrm{fresh}\left(t_{5}\right)$
(A10)${\mathrm{ME}\;\mathrm{believes}\;\mathrm{SN}}_{j}$
(A11)ME believes nonce
(A12)${\mathrm{ME}\;\mathrm{believes}\;M}_{\mathrm{id}}$
(A13)$\mathrm{ME}\;\mathrm{believes}\;\mathrm{fresh}\left(t_{7}\right)$
(A14)${\mathrm{ME}\;\mathrm{believes}\;\mathrm{IMD}\;\mathrm{Controls}\;K}_{\mathrm{ssk}}$
●    **Goals**(G1)GW believes nonce
(G2)${\mathrm{MD}\;\mathrm{believes}\;\mathrm{SN}}_{j}$
(G3)${\mathrm{MD}\;\mathrm{believes}\;M}_{\mathrm{id}}$
(G4)$\mathrm{ME}\;\mathrm{believes}\;\mathrm{IMD}\;\mathrm{believes}\;\mathrm{IMD}\overset{K_{\mathrm{ssk}}}{↔}\mathrm{ME}$
(G5)$\mathrm{IMD}\;\mathrm{believes}\;\mathrm{ME}\;\mathrm{believes}\;\mathrm{IMD}\overset{K_{\mathrm{ssk}}}{↔}\mathrm{ME}$
(G6)$\mathrm{IMD}\;\mathrm{believes}\;\mathrm{IMD}\overset{K_{\mathrm{ssk}}}{↔}\mathrm{ME}$
●    **Derivations**(D1)${\mathrm{GW}\;\mathrm{sees}\;\langle M{}^{\prime }}_{\mathrm{id}},\mathrm{nonce},{\;U}_{i},{\mathrm{SN}}_{j},t_{1}\rangle {}_{\mathrm{GW}\overset{K_{l}}{↔}\mathrm{ME}}\;$By (I1),DR(D2)$\mathrm{GW}\;\mathrm{believes}\;\mathrm{ME}\;\mathrm{said}\;\left[{M{}^{\prime }}_{\mathrm{id}},\mathrm{nonce},{\;U}_{i},{\mathrm{SN}}_{j},t_{1}\right]$By (D1),(A1),MM(D3)$\mathrm{GW}\;\mathrm{believes}\;\mathrm{ME}\;\mathrm{believes}\;\left[{M{}^{\prime }}_{\mathrm{id}},\mathrm{nonce},{\;U}_{i},{\mathrm{SN}}_{j},t_{1}\right]$By (D2),(A2),NV, FR(D4)GW believes ME believes nonceBy (D3),BC(D5)GW believes nonceBy (D4),(A3),JR(D6)${\mathrm{MD}\;\mathrm{sees}\;\langle M}_{\mathrm{id}},{\;U}_{i},{\mathrm{SN}}_{j},\mathrm{nonce},t_{3}\rangle {}_{\mathrm{GW}\overset{K_{\mathrm{GW}-U}}{↔}\mathrm{MD}}\;$(I2)(D7)$\mathrm{MD}\;\mathrm{believes}\;\mathrm{GW}\;\mathrm{said}\;\left[M_{\mathrm{id}},{\;U}_{i},{\mathrm{SN}}_{j},\mathrm{nonce},t_{3}\right]$By (D6), (A4), MM(D8)$\mathrm{MD}\;\mathrm{believes}\;\mathrm{GW}\;\mathrm{believes}\;\left[M_{\mathrm{id}},{\;U}_{i},{\mathrm{SN}}_{j},\mathrm{nonce},t_{3}\right]\;$By (D7), (A5), NV, FR(D9)${\mathrm{MD}\;\mathrm{believes}\;\mathrm{GW}\;\mathrm{believes}\;\mathrm{SN}}_{j}\;$By (D8), BC(D10)${\mathrm{MD}\;\mathrm{believes}\;\mathrm{SN}}_{j}$By (D9),(A6),JR(D11)${\mathrm{MD}\;\mathrm{believes}\;\mathrm{GW}\;\mathrm{believes}\;M}_{\mathrm{id}}$By (D8), BC(D12)${\mathrm{MD}\;\mathrm{believes}\;M}_{\mathrm{id}}$By (D11),(A7),JR(D13)${\mathrm{IMD}\;\mathrm{sees}\;\langle M}_{\mathrm{id}},{\;U}_{i},{\mathrm{SN}}_{j},\mathrm{nonce},t_{5}\rangle {}_{\mathrm{MD}\overset{K_{U-{\mathrm{SN}}_{j}}}{↔}\mathrm{IMD}}$(I3).(D14)$\mathrm{IMD}\;\mathrm{believes}\;\mathrm{MD}\;\mathrm{said}\;\left[M_{\mathrm{id}},{\;U}_{i},{\mathrm{SN}}_{j},\mathrm{nonce},t_{5}\right]\;$By (D13), (A8), MM(D15)$\mathrm{IMD}\;\mathrm{believes}\;\mathrm{MD}\;\mathrm{belives}\;\left[M_{\mathrm{id}},{\;U}_{i},{\mathrm{SN}}_{j},\mathrm{nonce},t_{5}\right]\;$By (D14), (A9), NV, FR(D16)${\mathrm{IMD}\;\mathrm{believes}\;\mathrm{MD}\;\mathrm{believes}\;M}_{\mathrm{id}}\;$By (D15), BC(D17)IMD believes MD believes nonceBy (D15), BC(D18)${\mathrm{ME}\;\mathrm{sees}\;\langle \mathrm{SN}}_{j},M_{\mathrm{id}},\;\mathrm{IMD}\overset{K_{\mathrm{ssk}}}{↔}\mathrm{ME},t_{7}\rangle {}_{\mathrm{IMD}\overset{K_{\mathrm{ssk}}}{↔}\mathrm{ME}}\;$By (I4)(D19)$\mathrm{ME}\;\mathrm{believes}\;\mathrm{IMD}\overset{K_{\mathrm{ssk}}}{↔}\mathrm{ME}\;$By (A10),(A11),(A12), BC(D20)$\mathrm{ME}\;\mathrm{believes}\;\mathrm{IMD}\;\mathrm{said}\;\left[{\mathrm{SN}}_{j},M_{\mathrm{id}},\;\mathrm{IMD}\overset{K_{\mathrm{ssk}}}{↔}\mathrm{ME},t_{7}\right]$By (D18), (D19), MM(D21)$\mathrm{ME}\;\mathrm{believes}\;\mathrm{IMD}\;\mathrm{belives}\;\left[{\mathrm{SN}}_{j},M_{\mathrm{id}},\;\mathrm{IMD}\overset{K_{\mathrm{ssk}}}{↔}\mathrm{ME},t_{7}\right]$By (D20), (A13), NV, FR(D22)$\mathrm{ME}\;\mathrm{believes}\;\mathrm{IMD}\;\mathrm{believes}\;\mathrm{IMD}\overset{K_{\mathrm{ssk}}}{↔}\mathrm{ME}$By (D21), BCAVISPA-based formal security analysis result.
Figure 11AVISPA result of Parvez et al.’s protocol.
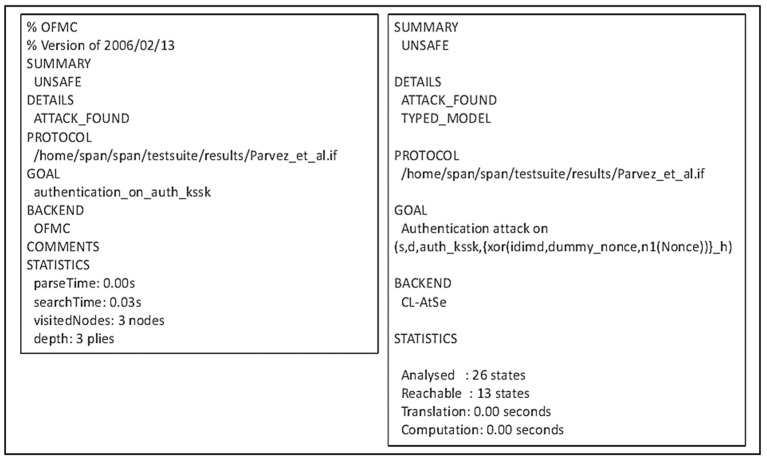



#### 4.3.5. Iqbal et al.’s Protocol

The proposed protocol [27] works between the sensor nodes (SN), controller (BS), and a medical server (MS). The SNs are (implanted) medical devices that sense vital physiological information. In this protocol, a BS is only used to assist the authentication process so that the SN directly communicates with the MS after successful authentication is achieved. The protocol is executed in three stages: deployment, authentication, and data communication, as shown in Figure 12 and Figure 13 presents the OFMC and CL-AtSe back-end results of the protocol.

BAN logic-based formal security analysis
●    **Idealization**(I1)$\mathrm{SN}\to \mathrm{BS}:\;{\left\{\mathrm{Nonce},\mathrm{SK},{\mathrm{ID}}_{i}\right\}}_{M_{\mathrm{sk}}}$
(I2)$\mathrm{BS}\to \mathrm{SN}:\;{\left\{\mathrm{Nonce}\right\}}_{\mathrm{SK}}$
(I3)$\mathrm{SN}\to \mathrm{MS}:\;{\left\{\mathrm{SN}\overset{\mathrm{SK}}{↔}\mathrm{MS}\right\}}_{M_{\mathrm{sk}}}$
(I4)$\mathrm{SN}\to \mathrm{MS}:\;{\left\{M\right\}}_{\mathrm{SK}}$
●    **Assumption**(A1)$\mathrm{BS}\;\mathrm{believes}\;\mathrm{SN}\overset{M_{\mathrm{sk}}}{↔}\mathrm{BS}$
(A2)BS believes fresh(Nonce)
(A3)BS believes SN Controls SK
(A4)SN believes SK
(A5)SN believes fresh(Nonce)
(A6)$\mathrm{MS}\;\mathrm{believes}\;\mathrm{MS}\overset{M_{\mathrm{sk}}}{↔}\mathrm{SN}$
●    **Hypotheses**(H1)$\mathrm{MS}\;\mathrm{believes}\;\mathrm{fresh}\left(\mathrm{MS}\overset{\mathrm{SK}}{↔}\mathrm{BS}\right)$
(H2)$\mathrm{MS}\;\mathrm{believes}\;\mathrm{SN}\;\mathrm{Controls}\;\mathrm{MS}\overset{\mathrm{SK}}{↔}\mathrm{BS}$
●    **Goals**(G1)$\mathrm{BS}\;\mathrm{believes}\;\mathrm{SN}\;\mathrm{believes}\;\mathrm{SN}\overset{\mathrm{SK}}{↔}\mathrm{BS}$
(G2)$\mathrm{BS}\;\mathrm{believes}\;\mathrm{SN}\overset{\mathrm{SK}}{↔}\mathrm{BS}$
(G3)$\mathrm{SN}\;\mathrm{believes}\;\mathrm{BS}\;\mathrm{believes}\;\mathrm{SN}\overset{\mathrm{SK}}{↔}\mathrm{BS}$
(G4)$\mathrm{MS}\;\mathrm{believes}\;\mathrm{SN}\;\mathrm{believes}\;\mathrm{SN}\overset{\mathrm{SK}}{↔}\mathrm{MS}$
(G5)$\mathrm{MS}\;\mathrm{believes}\;\mathrm{SN}\overset{\mathrm{SK}}{↔}\mathrm{MS}$
●    **Derivations**(D1)$\mathrm{BS}\;\mathrm{sees}\;\langle \mathrm{Nonce},\mathrm{SK},{\;\mathrm{ID}}_{i}\rangle {}_{M_{\mathrm{SK}}}\;$(I1)(D2)$\mathrm{BS}\;\mathrm{believes}\;\mathrm{SN}\;\mathrm{said}\;\left[\mathrm{Nonce},\mathrm{SK},{\;\mathrm{ID}}_{i}\right]$By (D1),(A1),MM(D3)$\mathrm{BS}\;\mathrm{believes}\;\mathrm{SN}\;\mathrm{believes}\;\left[\mathrm{Nonce},\mathrm{SK},{\;\mathrm{ID}}_{i}\right]$By (D2),(A2),NV, FR(D4)BS believes SN believes SK By (D3),BC(D5)BS believes SK By (D4),(A3), JR(D6)$\mathrm{SN}\;\mathrm{sees}\;\langle \mathrm{Nonce},\mathrm{SN}\overset{\mathrm{SK}}{↔}\mathrm{BS}\rangle {}_{\mathrm{SK}}$(I2)(D7)$\mathrm{SN}\;\mathrm{believes}\;\mathrm{BS}\;\mathrm{said}\;\left[\mathrm{Nonce},\mathrm{SN}\overset{\mathrm{SK}}{↔}\mathrm{BS}\right]$By (D6),(A4),MM(D8)$\mathrm{SN}\;\mathrm{believes}\;\mathrm{BS}\;\mathrm{believes}\;\mathrm{SN}\overset{\mathrm{SK}}{↔}\mathrm{BS}\;$By (D7),(A5),NV, FR, BC(D9)$\mathrm{MS}\;\mathrm{sees}\;\langle \mathrm{SN}\overset{\mathrm{SK}}{↔}\mathrm{MS}\rangle {}_{M_{\mathrm{SK}}}$(I3)(D10)$\mathrm{MS}\;\mathrm{believes}\;\mathrm{SN}\;\mathrm{said}\;\left[\mathrm{SN}\overset{\mathrm{SK}}{↔}\mathrm{MS}\right]$By (D9),(A6),MM(D11)$\mathrm{MS}\;\mathrm{believes}\;\mathrm{SN}\;\mathrm{believes}\;\mathrm{SN}\overset{\mathrm{SK}}{↔}\mathrm{BS}\;$By (D10),(H1),NV, FR(D12)$\mathrm{MS}\;\mathrm{believes}\;\mathrm{SN}\overset{\mathrm{SK}}{↔}\mathrm{BS}\;$By (D10), (H2), JRAVISPA-based formal security analysis result.
Figure 13AVISPA result of Iqbal et al.’s protocol.
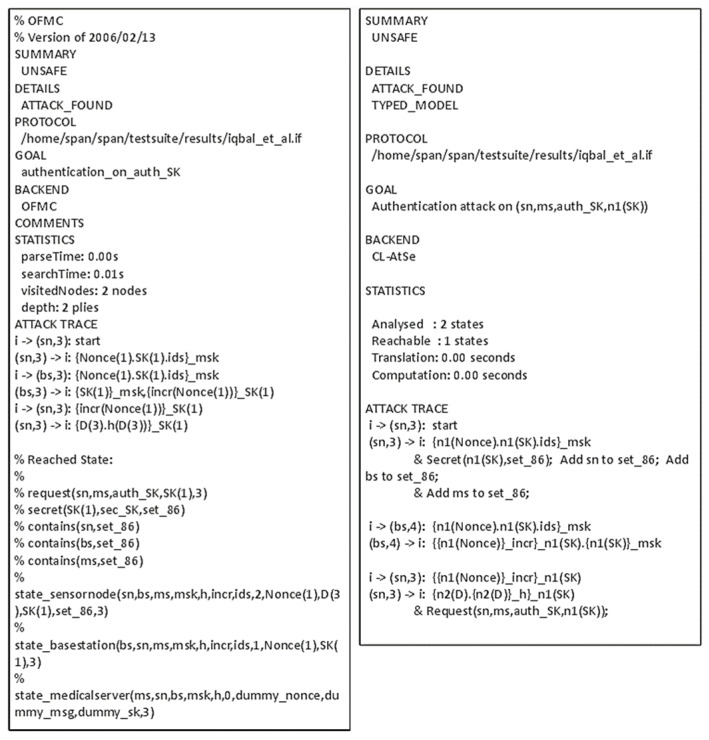



#### 4.3.6. He and Zeadally’s Protocol

He and Zeadally’s authentication protocol [28] comprises a programmer/controller, the AAL server, and a user. The controller is responsible for communicating with the IMDs and receiving collected physiological information. Once such information is collected, it can be accessed by a remote user after the AAL server authenticates the user. Furthermore, the controller may communicate with different devices, such as home robots for immediate nearby assistance, located in the patient’s premises. The protocol is also executed in two stages: registration and authentication, as shown in Figure 14 and Figure 15 illustrates the OFMC and CL-AtSe back-end results of the protocol.

BAN logic-based formal security analysis.
●    **Idealization**(I1)$U\to A:\;{\left\{{\mathrm{ID}}_{U},{\mathrm{ID}}_{C},\overset{P_{U}}{\to }U,{\mathrm{TS}}_{U},A\overset{{\mathrm{TK}}_{A-U}}{↔}U\right\}}_{A\overset{{\mathrm{TK}}_{A-U}}{↔}U}\langle \overset{P_{U}}{\to }U,{\mathrm{TS}}_{U},A\overset{K_{A-U}}{↔}U\rangle {}_{A\overset{K_{A-U}}{↔}U}$(I2)$A\to C:\;{\left\{{\mathrm{ID}}_{U},{\mathrm{ID}}_{C},\overset{P_{U}}{\to }U,{\mathrm{TS}}_{A_{1}},A\overset{K_{A-C}}{↔}C\right\}}_{A\overset{K_{A-C}}{↔}C}$(I3)$C\to A:\;{\left\{{\mathrm{ID}}_{U},{\mathrm{ID}}_{C},\overset{P_{C}}{\to }C,{\mathrm{TS}}_{C},A\overset{K_{A-C}}{↔}C\right\}}_{A\overset{K_{A-C}}{↔}C}$(I4)$A\to U:\;{\left\{{\mathrm{ID}}_{U},{\mathrm{ID}}_{C},\overset{P_{C}}{\to }C,{\mathrm{TS}}_{A_{2}},A\overset{{\mathrm{TK}}_{A-U}}{↔}U\right\}}_{A\overset{{\mathrm{TK}}_{A-U}}{↔}U}$●    **Assumption**(A1)$A\;\mathrm{believes}\;A\overset{K_{A-U}}{↔}U$
(A2)$A\;\mathrm{believes}\;\#\left({\mathrm{TS}}_{U}\right)$
(A3)$A\;\mathrm{believes}\;\overset{Q_{A}}{\to }A$
(A4)$C\;\mathrm{believes}\;A\overset{K_{A-C}}{↔}C$
(A5)$C\;\mathrm{believes}\;\#\left({\mathrm{TS}}_{A_{1}}\right)$
(A6)$C\;\mathrm{believes}\;\overset{Q_{C}}{\to }C$
(A7)$A\;\mathrm{believes}A\overset{K_{A-C}}{↔}C$
(A8)$A\;\mathrm{believes}\;\#\left({\mathrm{TS}}_{C}\right)$
(A9)$U\;\mathrm{believes}\;A\overset{{\mathrm{TK}}_{A-U}}{↔}U$
(A10)$U\;\mathrm{believes}\;\#\left({\mathrm{TS}}_{A_{2}}\right)$
(A11)$U\;\mathrm{believes}\;\overset{Q_{U}}{\to }U$

Figure 14He and Zeadally’s protocol. (**a**) Registration phase. (**b**) Authentication phase.
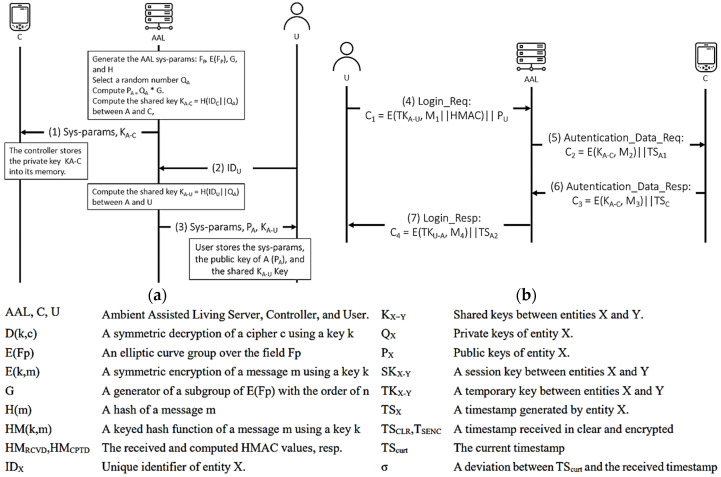

●    **Hypotheses**(H1)$A\;\mathrm{believes}\;U\;\mathrm{Controls}\overset{P_{U}}{\to }U\;$
●    **Goals**(G1)$A\;\mathrm{believes}\;A\overset{{\mathrm{TK}}_{A-U}}{↔}U$
(G2)$A\;\mathrm{believes}\;U\;\mathrm{believes}\;A\overset{K_{A-U}}{↔}U$
(G3)${A\;\mathrm{believes}\;U\;\mathrm{believes}\;\mathrm{ID}}_{U}$
(G4)$A\;\mathrm{believes}\;U\;\mathrm{believes}\;A\overset{{\mathrm{TK}}_{A-U}}{↔}U$
(G5)${C\;\mathrm{believes}\;A\;\mathrm{believes}\;\mathrm{ID}}_{U}$
(G6)$C\;\mathrm{believes}\;C\overset{{\mathrm{SK}}_{C-U}}{↔}U$
(G7)$C\;\mathrm{believes}\;A\;\mathrm{believes}\;C\overset{K_{A-C}}{↔}A$
(G8)${A\;\mathrm{believes}\;C\;\mathrm{believes}\;\mathrm{ID}}_{C}$
(G9)$A\;\mathrm{believes}\;C\;\mathrm{believes}\;C\overset{K_{A-C}}{↔}A$
(G10)${U\;\mathrm{believes}\;A\;\mathrm{believes}\;\mathrm{ID}}_{C}$
(G11)$U\;\mathrm{believes}\;C\overset{{\mathrm{SK}}_{C-U}}{↔}U$
(G12)$U\;\mathrm{believes}\;A\;\mathrm{believes}\;U\overset{{\mathrm{TK}}_{A-U}}{↔}A$
●    **Derivations**(D1)${A\;\mathrm{sees}\;\langle \mathrm{ID}}_{U},{\mathrm{ID}}_{C},{\mathrm{TS}}_{U},A\overset{{\mathrm{TK}}_{A-U}}{↔}U\rangle {}_{A\overset{{\mathrm{TK}}_{A-U}}{↔}U}$, $\langle \overset{P_{U}}{\to }U,{\mathrm{TS}}_{U},A\overset{K_{A-U}}{↔}U\rangle {}_{A\overset{K_{A-U}}{↔}U}$(I1)(D2)$A\;\mathrm{believes}\;U\;\mathrm{said}\;\left[\overset{P_{U}}{\to }U,{\mathrm{TS}}_{U},A\overset{K_{A-U}}{↔}U\right]\;$By (D1),(A1),MM(D3)$A\;\mathrm{believes}\;U\;\mathrm{believes}\;\left[\overset{P_{U}}{\to }U,{\mathrm{TS}}_{U},A\overset{K_{A-U}}{↔}U\right]$By (D2),(A2),FR,NV(D4)$A\;\mathrm{believes}\;U\;\mathrm{believes}\;\overset{P_{U}}{\to }U\;$By (D3),BC(D5)$A\;\mathrm{believes}\;\overset{P_{U}}{\to }U\;$By (D4),(H1), JR(D6)$A\;\mathrm{believes}\;A\overset{{\mathrm{TK}}_{A-U}}{↔}U\;$By (D5),(A3),BC(D7)$A\;\mathrm{believes}\;U\;\mathrm{believes}\;A\overset{K_{A-U}}{↔}U\;$By (D3),BC(D8)$A\;\mathrm{believes}\;U\;\mathrm{said}\;\left[{\mathrm{ID}}_{U},{\mathrm{ID}}_{C},{\mathrm{TS}}_{U},A\overset{{\mathrm{TK}}_{A-U}}{↔}U\right]$By (D1),(D6),MM(D9)$A\;\mathrm{believes}\;U\;\mathrm{believes}\;\left[{\mathrm{ID}}_{U},{\mathrm{ID}}_{C},{\mathrm{TS}}_{U},A\overset{{\mathrm{TK}}_{A-U}}{↔}U\right]\;$By (D8),(A2),FR,NV(D10)${A\;\mathrm{believes}\;U\;\mathrm{believes}\;\mathrm{ID}}_{U}\;$By (D9),BC(D11)$A\;\mathrm{believes}\;U\;\mathrm{believes}\;U\overset{{\mathrm{TK}}_{A-U}}{↔}A\;$By (D9),BC(D12)$C\;\mathrm{sees}\;{\left\{{\mathrm{ID}}_{U},{\mathrm{ID}}_{C},\overset{P_{U}}{\to }U,{\mathrm{TS}}_{A},A\overset{K_{A-C}}{↔}C\right\}}_{A\overset{K_{A-C}}{↔}C}$(I2)(D13)$C\;\mathrm{believes}\;A\;\mathrm{said}\;\left[{\mathrm{ID}}_{U},{\mathrm{ID}}_{C},\overset{P_{U}}{\to }U,{\mathrm{TS}}_{A},A\overset{K_{A-C}}{↔}C\right]$By (D12),(A4),MM(D14)$C\;\mathrm{believes}\;A\;\mathrm{believes}\;\left[{\mathrm{ID}}_{U},{\mathrm{ID}}_{C},\overset{P_{U}}{\to }U,{\mathrm{TS}}_{A},A\overset{K_{A-C}}{↔}C\right]$By (D13),(A5),FR,NV(D15)${C\;\mathrm{believes}\;A\;\mathrm{believes}\;\mathrm{ID}}_{U}\;$By (D14),BC(D16)$C\;\mathrm{believes}\;C\overset{{\mathrm{SK}}_{C-U}}{↔}U\;$
By (D13), BC, (A6), DH
(D17)$C\;\mathrm{believes}\;A\;\mathrm{believes}\;C\overset{K_{A-C}}{↔}A\;$By (D14),BC(D18)$A\;\mathrm{sees}\;{\left\{{\mathrm{ID}}_{U},{\mathrm{ID}}_{C},\overset{P_{C}}{\to }C,{\mathrm{TS}}_{C},A\overset{K_{A-C}}{↔}C\right\}}_{A\overset{K_{A-C}}{↔}C}$(I3)(D19)$A\;\mathrm{believes}\;C\;\mathrm{said}\;\left[{\mathrm{ID}}_{U},{\mathrm{ID}}_{C},\overset{P_{C}}{\to }C,{\mathrm{TS}}_{C},A\overset{K_{A-C}}{↔}C\right]$By (D18),(A7),MM(D20)$A\;\mathrm{believes}\;C\;\mathrm{believes}\;\left[{\mathrm{ID}}_{U},{\mathrm{ID}}_{C},\overset{P_{C}}{\to }C,{\mathrm{TS}}_{C},A\overset{K_{A-C}}{↔}C\right]$By (D19),(A8),FR,NV(D21)${A\;\mathrm{believes}\;C\;\mathrm{believes}\;\mathrm{ID}}_{C}\;$By (D20),BC(D22)$A\;\mathrm{believes}\;C\;\mathrm{believes}\;C\overset{K_{A-C}}{↔}A\;$By (D20),BC(D23)$U\;\mathrm{sees}\;{\left\{{\mathrm{ID}}_{U},{\mathrm{ID}}_{C},\overset{P_{C}}{\to }C,{\mathrm{TS}}_{A_{2}},A\overset{{\mathrm{TK}}_{A-U}}{↔}U\right\}}_{A\overset{{\mathrm{TK}}_{A-U}}{↔}U}$(I4)(D24)$U\;\mathrm{believes}\;A\;\mathrm{said}\;\left[{\mathrm{ID}}_{U},{\mathrm{ID}}_{C},\overset{P_{C}}{\to }C,{\mathrm{TS}}_{A_{2}},A\overset{{\mathrm{TK}}_{A-U}}{↔}U\right]$By (D23),(A9),MM(D25)$U\;\mathrm{believes}\;A\;\mathrm{believes}\;\left[{\mathrm{ID}}_{U},{\mathrm{ID}}_{C},\overset{P_{C}}{\to }C,{\mathrm{TS}}_{A_{2}},A\overset{{\mathrm{TK}}_{A-U}}{↔}U\right]$By (D24),(A10),FR,NV(D26)${U\;\mathrm{believes}\;A\;\mathrm{believes}\;\mathrm{ID}}_{C}\;$By (D25),BC(D27)$U\;\mathrm{believes}\;C\overset{{\mathrm{SK}}_{C-U}}{↔}U\;$By (D24),BC,(A11),DH(D28)$U\;\mathrm{believes}\;A\;\mathrm{believes}\;A\overset{{\mathrm{TK}}_{A-U}}{↔}U\;$By (D25),BCAVISPA-based formal security analysis result.
Figure 15AVISPA result of He and Zeadally’s protocol.
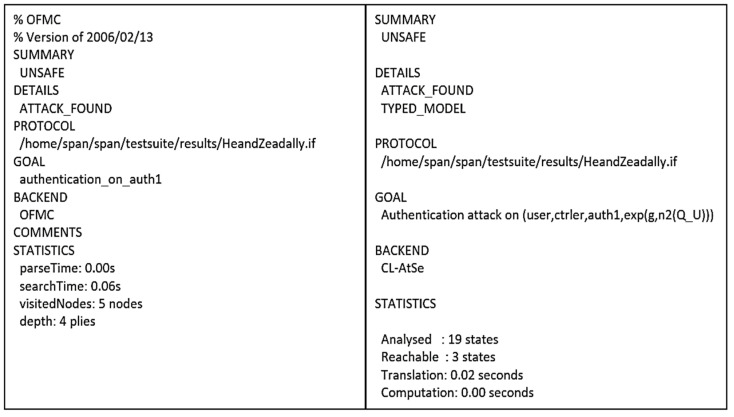



#### 4.3.7. Ellouze et al.’s Protocol

This protocol [29] is a mutual authentication protocol for cardiac IMDs that integrates a powerless device called wireless identification and sensing platform (WISP) with IMDs to conserve the battery lifetime of IMDs by drawing energy from an RFID reader. The authentication scheme operates in regular and emergency modes between the WISP and the RFID reader. The final goal is to create mutual authentication between the programmer and the IMD. Figure 16 shows both modes of the protocol. The authors of this protocol performed AVISPA-based security verification and claimed that the protocol is secure. Hence, only BAN logic-based analysis is performed here.

BAN logic-based formal security analysis.Regular mode.
●    **Idealization**(I1)$R\to W:\langle {\;N}_{R},{\;\mathrm{ID}}_{R},W\overset{K}{↔}R\rangle {}_{W\overset{K}{↔}R}$(I2)$W\to R:\langle {\;N}_{R},{\;N}_{W},{\mathrm{ID}}_{W},W\overset{K}{↔}R\rangle {}_{W\overset{K}{↔}R}$(I3)$R\to \mathrm{WISP}:\langle {\;N}_{W},{\mathrm{Seq}}_{1},W\overset{K‘}{↔}R\rangle {}_{W\overset{K‘}{↔}R}$●    **Assumption**(A1)$W\;\mathrm{believes}\;W\overset{K}{↔}R$
(A2)$W\;\mathrm{believes}\;\#\left(N_{W}\right)$
(A3)$R\;\mathrm{believes}\;W\overset{K}{↔}R$
(A4)$R\;\mathrm{believes}\;\#\left(N_{R}\right)$
●    **Hypotheses**(H1)$W\;\mathrm{believes}\;\#\left(N_{R}\right)$
(H2)$R\;\mathrm{believes}\;\#\left(N_{W}\right)$
●    **Goals**(G1)${W\;\mathrm{believes}\;R\;\mathrm{believes}\;\mathrm{ID}}_{R}$
(G2)$W\;\mathrm{believes}\;R\;\mathrm{believes}\;W\overset{K}{↔}R$
(G3)$W\;\mathrm{believes}W\overset{K‘}{↔}R$
(G4)${R\;\mathrm{believes}\;W\;\mathrm{believes}\;\mathrm{ID}}_{W}$
(G5)$R\;\mathrm{believes}\;W\;\mathrm{believes}\;W\overset{K}{↔}R$
(G6)$R\;\mathrm{believes}\;W\overset{K‘}{↔}R$
(G7)$W\;\mathrm{believes}\;R\;\mathrm{believes}\;W\overset{K‘}{↔}R$
●    **Derivations**(D1)${W\;\mathrm{sees}\;\langle N}_{R},{\;\mathrm{ID}}_{R},W\overset{K}{↔}R\rangle {}_{W\overset{K}{↔}R}$(I1)(D2)$W\;\mathrm{believes}\;R\;\mathrm{said}\;\left[N_{R},{\;\mathrm{ID}}_{R},W\overset{K}{↔}R\right]$By (D1),(A1),MM(D3)${W\;\mathrm{believes}\;R\;\mathrm{believes}\;\mathrm{ID}}_{R}\;$By (D2),(H1),FR,NV(D4)$W\;\mathrm{believes}\;R\;\mathrm{believes}\;W\overset{K}{↔}R$By (D2),(H1),FR,NV(D5)$W\;\mathrm{believes}\;W\overset{K‘}{↔}R\;$By (A1),(A2),BC(D6)${R\;\mathrm{sees}\;\langle N}_{R},{\;N}_{W},{\mathrm{ID}}_{W},W\overset{K}{↔}R\rangle {}_{W\overset{K}{↔}R}\;$(I2)(D7)$R\;\mathrm{believes}\;W\;\mathrm{said}\;\left[N_{R},{\;N}_{W},{\mathrm{ID}}_{W},W\overset{K}{↔}R\right]\;$By (D6),(A3),MM(D8)${R\;\mathrm{believes}\;W\;\mathrm{believes}\;\mathrm{ID}}_{W}\;$By (D7),(A4),FR,NV(D9)$R\;\mathrm{believes}\;W\;\mathrm{believes}\;W\overset{K}{↔}R$By (D7),(A4),FR,NV(D10)$R\;\mathrm{believes}\;W\overset{K‘}{↔}R\;$By (A3),(H2),BC(D11)${W\;\mathrm{sees}\;\langle N}_{W},{\mathrm{Seq}}_{1},W\overset{K‘}{↔}R\rangle {}_{W\overset{K‘}{↔}R}\;$By (D8), BC(D12)$W\;\mathrm{believes}\;R\;\mathrm{said}\;\left[N_{W},{\mathrm{Seq}}_{1},W\overset{K‘}{↔}R\right]\;$By (D11),(D5),MM(D13)$W\;\mathrm{believes}\;R\;\mathrm{believes}\;W\overset{K‘}{↔}R\;$
By (D12), (A2), FR, NV
Emergency Mode
●    **Idealization**(I1)$R\to W:\langle \;Q,N_{R},{\mathrm{ID}}_{R},W\overset{K_{\mathrm{bio}}}{↔}R\rangle {}_{W\overset{K_{\mathrm{bio}}}{↔}R}$(I2)$W\to R:\langle {\;N}_{R},{\;N}_{W},{\mathrm{ID}}_{W},W\overset{K_{\mathrm{bio}}}{↔}R\rangle {}_{W\overset{K_{\mathrm{bio}}}{↔}R}$(I3)$R\to W:\langle {\;N}_{W},{\mathrm{Seq}}_{1},W\overset{K‘}{↔}R\rangle {}_{W\overset{K‘}{↔}R}$
Figure 16Ellouze et al.’s protocol. (**a**) Regular Mode. (**b**) Emergency Mode.
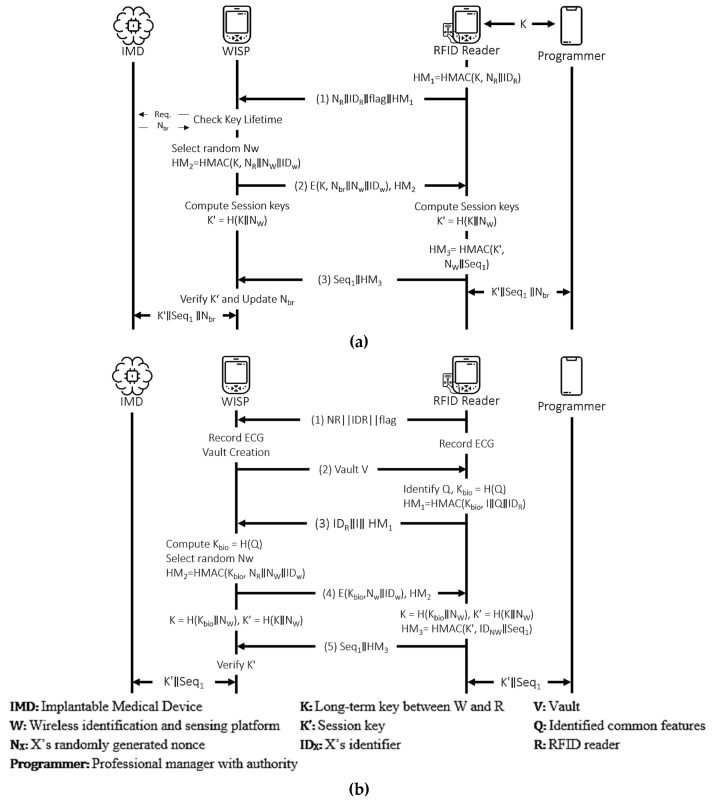

●    **Assumption**(A1)$W\;\mathrm{believes}\;W\overset{K_{\mathrm{bio}}}{↔}R$
(A2)$W\;\mathrm{believes}\;\#\left(N_{W}\right)$
(A3)$R\;\mathrm{believes}\;W\overset{K_{\mathrm{bio}}}{↔}R$
(A4)$R\;\mathrm{believes}\;\#\left(N_{R}\right)$
●    **Hypotheses**(H1)$W\;\mathrm{believes}\;\#\left(N_{R}\right)$
(H2)$R\;\mathrm{believes}\;\#\left(N_{W}\right)$
●    **Goals**(G1)${W\;\mathrm{believes}\;R\;\mathrm{believes}\;\mathrm{ID}}_{R}$
(G2)$W\;\mathrm{believes}\;R\;\mathrm{believes}\;W\overset{K_{\mathrm{bio}}}{↔}R$
(G3)$W\;\mathrm{believes}W\overset{K‘}{↔}R$
(G4)${R\;\mathrm{believes}\;W\;\mathrm{believes}\;\mathrm{ID}}_{W}$
(G5)$R\;\mathrm{believes}\;W\;\mathrm{believes}\;W\overset{K_{\mathrm{bio}}}{↔}R$
(G6)$R\;\mathrm{believes}\;W\overset{K‘}{↔}R$
(G7)$W\;\mathrm{believes}\;R\;\mathrm{believes}\;W\overset{K‘}{↔}R$
●    **Derivations**(D1)$W\;\mathrm{sees}\;\langle Q,N_{R},{\mathrm{ID}}_{R},W\overset{K_{\mathrm{bio}}}{↔}R\rangle {}_{W\overset{K_{\mathrm{bio}}}{↔}R}\;$(I1)(D2)W believes R said. $\left[Q,N_{R},{\mathrm{ID}}_{R},W\overset{K_{\mathrm{bio}}}{↔}R\right]\;$By (D1),(A1),MM(D3)${W\;\mathrm{believes}\;R\;\mathrm{believes}\;\mathrm{ID}}_{R}$. By (D2),(H1),FR,NV(D4)$W\;\mathrm{believes}\;R\;\mathrm{believes}\;W\overset{K_{\mathrm{bio}}}{↔}R$By (D2),(H1),FR,NV(D5)$W\;\mathrm{believes}\;W\overset{K‘}{↔}R$By (A1),(A2),BC(D6)${R\;\mathrm{sees}\;\langle N}_{R},{\;N}_{W},{\mathrm{ID}}_{W},W\overset{K_{\mathrm{bio}}}{↔}R\rangle {}_{W\overset{K_{\mathrm{bio}}}{↔}R}$(I2)(D7)R believes W said $\left[N_{R},NW,{\mathrm{ID}}_{W},W\overset{K_{\mathrm{bio}}}{↔}R\right]$By (D6),(A3),MM(D8)R believes W believes ${\mathrm{ID}}_{W}\;$BY (D7),(A4),FR,NV(D9)R believes W believes $W\overset{K_{\mathrm{bio}}}{↔}R\;$By (D7),(A4),FR,NV(D10)$R\;\mathrm{believes}\;W\overset{K‘}{↔}R$By (A3),(H2),BC(D11)${W\;\mathrm{sees}\;\langle N}_{W},{\mathrm{Seq}}_{1},W\overset{K‘}{↔}R\rangle {}_{W\overset{K‘}{↔}R}\;$(I3)(D12)W believes R said $\left[N_{W},{\mathrm{Seq}}_{1},W\overset{K‘}{↔}R\right]$By (D11),(D5),MM(D13)W believes R believes $W\overset{K‘}{↔}R$By (D12),(A2),FR,NV

## 5. Discussion

The authentication protocols described and analyzed in the previous section have shown the importance of formally analyzing security protocols for usage reliability.

Khan et al.’s protocol is safe as per the output of both BAN logic and AVISPA in satisfying the goals. The hub node can be sure about the validity of the temporary identification (G1 and G2) and the auxiliary authentication parameter (G3 and G4). Furthermore, the sensor node trusts the newly generated session key (G6 and G7).

The proxy-assisted access control scheme proposed by Wu et al. is the second safe protocol for the authentication goals set. The main objective of the protocol is to device a shared symmetric key K_t_ that the programmer and the IMD will use to secure the information exchanged. Accordingly, (G2) to (G5), (G8), and (G9) show that this objective is satisfied. Furthermore, other less essential facts that involve the IDs of the participating agents are authentic.

Chi et al.’s access control scheme with forensic capability is designed to safeguard IMDs from unauthorized access. The protocol is secure as per the results of BAN logic and AVISPA on authenticity and secrecy properties. The goals in the BAN logic analysis investigated the authentication between the IMD, smartphone, and the programmer through the keys K_d_, K_r_, K_p_, and K_i_.

The user authentication scheme in WBAN, as proposed by Parvez et al., is found to be unsafe, by both AVISPA and BAN logic, on the authentication of the shared key K_ssk_. The shared key that will be used by the medical expert and the IMD is computed from the nonce, SN_j_, and M_id_. In terms of BAN logic, this means the IMD has to believe these values to believe the computed session key. Consequently, the derivations (D11) to (D13) alone cannot enable the IMD to derive its belief to the shared key- which calls for two new hypotheses about the control of the nonce and M_id_ by the mobile device that acts as a proxy between the IMD and the external devices. Such hypotheses may not be accurate given that ME and GW generated these values, respectively. However, since the message passed from the MD to the IMD is fresh and protected by the key that both parties trust, we may still be convinced that the IMD can derive the session key. The final goal, (G5), can be derived after a new message arrives from the ME to IMD using the session key K_ssk_.

Iqbal et al.’s authentication and key agreement scheme are proposed for node authentication in the body sensor environment. The protocol has some serious issues, typically concerning reply attacks. Specifically, the security goals (G4) and (G5) related to the mutual authentication between the medical server and the IMD cannot be satisfied as-is. The medical server cannot be sure about the freshness of the shared session key forwarded by the base station, making the message vulnerable to replay attack. Consequently, the hypotheses (H1) and (H2) need to be added to maintain authentication. More importantly, it is possible to improve the protocol by including a nonce along with the session key SK when BS sends the message to the MS.

He and Zeadally’s scheme aims to improve the security of ambient assisted living. It mainly focuses on the mutual authentication between the Controller and the User via the AAL server. With this regard, the goals (G3), (G5), (G8), and (G10) refer to the secure information exchange, while (G6) and (G11) specify the secure session key exchange between the User and the Controller. The remaining goals are related to the exchange of symmetric keys among all the participants of the authentication scheme. The result of both the BAN logic and AVISPA illustrate that it is not possible to conclusively state the protocol as safe to use. That is, the derivations show that for the AAL server to believe that TK_A-U_ is a key that is only known by itself and the User (G1), it must first believe that PU is the public key of the User that is encrypting the messages by the key TK_A-U_ (G2). This, in turn, needs the AAL server to believe that this User has jurisdiction over the public key PU, meaning that the AAL server has to trust this User concerning PU (H1). Consequently, we cannot prove the goals (G1) and (G2) without the hypothesis we added.

Ellouze et al.’s scheme is a specific authentication protocol proposed for cardiac IMDs with powerless authentication mechanisms. The protocol operates in both emergency and regular modes to authenticate the programmer to the IMD and vice versa. The authors of this protocol have performed AVISPA based formal security analysis and reported that the protocol is safe. However, when the protocol is analyzed using BAN logic, a contrary result is found. The result from the analysis of the BAN logic in the emergency mode of the protocol shows the requirement of two additional hypotheses to satisfy the authentication between the WISP and the RFID Reader. Specifically, the WISP cannot conclusively believe the K_Bio_. This key will be used to derive the session key K’ latter if the reader believes it without guaranteeing the freshness of NR. Furthermore, the security goal that conditions the guarantee for the WISP that the RFID Reader believes the session key K’ (G7) can only be satisfied if the freshness of NW is guaranteed. Concerning the regular mode, the same issue exists as shown in the hypotheses (H1) and (H2) for the derivation of (D3), (D4), and (D10).

## 6. Comparative Analysis

### 6.1. Comparison by Security Strength

Here, we compare the authentication schemes that are formally analyzed in Section 3. The comparisons are based on security properties, key features that IMD authentication protocols need to possess, computational overhead, and latency. Accordingly, each of the authentication protocols is checked against different security requirements (integrity (INT), confidentiality (CNF), authentication (AUT), session key agreement (SKA), perfect forward secrecy (PFS), and replay attack protection (RAP)) as shown in Table 3.

### 6.2. Comparison by Functionality

Various vital functionalities are expected to be satisfied by authentication protocols, in particular features like emergency authentication (EMA), key update mechanisms (KUM), adaptability (ADP), application (APP), and anonymity (ANO) is used to compare the protocols. The comparison result of the authentication schemes concerning these functionalities is shown in Table 4.

### 6.3. Comparison by Computational and Communicational Overhead

The computational and communication overheads, in terms of time, to perform the cryptographic operations (such as the number of signatures, symmetric, and asymmetric key encryption and decryption, hash functions, and XOR operations) [46,47,48], and size of the messages communicated [46,49], respectively, are shown in Table 5, Table 6, Table 7 and Table 8. The comparison of protocols concerning computational and communication overheads is depicted in Figure 17.

### 6.4. Overall Comparison of the Authentication Protocols

The comparison metrics—security strength, functionality, and efficiency—can be collectively used to understand better these schemes regarding security, competence, and capability. Such comparison can be best described in a triangular graph, as shown in Figure 18.

Figure 18 shows that Khan et al.’s scheme is located at the center since the protocol satisfies all the three metrics compared to the other protocols. On the other hand, while Iqbal et al. and Ellouze et al.’s schemes are only good at efficiency, Wu et al.’s and He and Zeadally’s protocols fulfill only security and functionality, respectively. Concerning Chi et al.’s scheme, only functionality and security is satisfied while efficiency is not met. On the other hand, the Parvez et al.’s, satisfy functionality and efficiency while falling short in meeting security.

## 7. Conclusions

In this research, we studied various IMD-related security and privacy requirements, such as confidentiality, integrity, availability, mutual authentication, non-traceability, user anonymity, session-key agreement, forward and backward secrecy, known attack resistance, device-existence privacy, device-type privacy, specific-device ID privacy, measurement and log privacy, and bearer privacy. Furthermore, we examined some of the well-known threats of IMDs: learning the existence of IMD, eavesdropping on the wireless channel that links the IMDs to the external devices, replay attacks by forwarding exchanged messages at a later time, changing critical settings of the implants by producing new commands, and exhausting the battery life of IMDs to execute denial of service attacks. After studying various IMD-related security and privacy concepts, we have used a formal approach to test the strength of seven contemporary authentication schemes designed to thwart attacks surrounding IMD-enabled systems. Consequently, we formally analyzed these authentication schemes using AVISPA and BAN logic, and compared them against their security strength, computational and communication overheads, and other features. The result analysis indicates that Khan et al. is the lightest and fastest while preserving privacy and satisfying the security properties shown in Table 3. The protocol uses only a cryptographic hash function and a bitwise XOR function, which made its computational and communication overheads lighter. Furthermore, the protocol is adaptable with minimal effort for the already implanted devices and no trouble for the yet-to-be implanted devices. Another important lesson taken from the analysis of the protocols is the necessity of formal security verification before IMD protocols are released for public use. In addition, IMD authentication schemes need to satisfy essential functionalities such as portability and emergency authentication while remaining lightweight. Accordingly, there is an interest to design a new security protocol for IMD-enabled insulin pumps in the future, which will serve as an artificial pancreas for patients in need. While designing such protocols, the authors would like to apply the essential lessons learned during this study. The newly designed protocol should be formally analyzed while satisfying the emergency authentication, adaptability, key update mechanisms, and anonymity requirements. The authors would also put forth an effort to balance these requirements with efficient communication and computational overhead and good attack resistance.

## Figures and Tables

**Figure 1 sensors-21-08383-f001:**
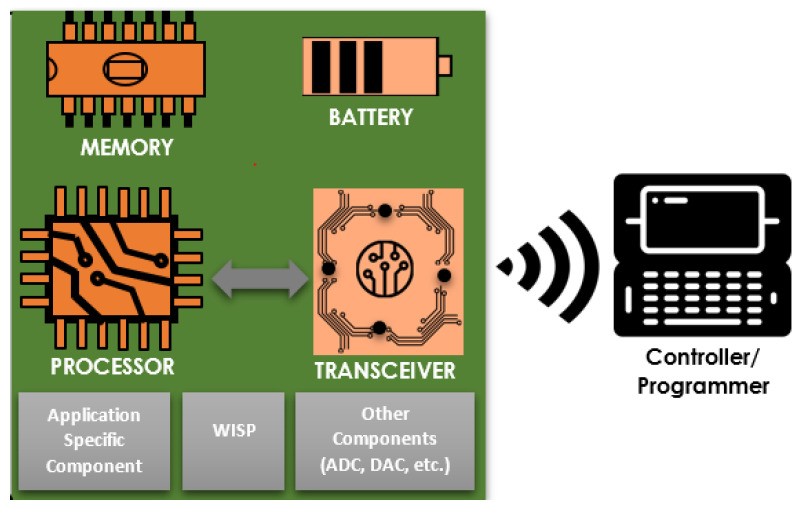
A typical IMD system architecture.

**Figure 2 sensors-21-08383-f002:**
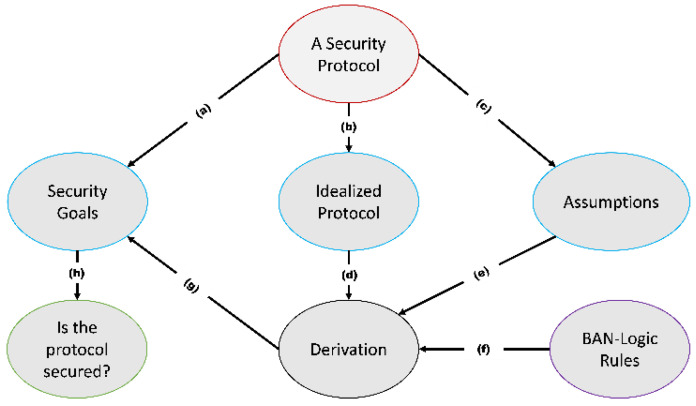
A typical BAN logic formal analysis procedure. (**a**) Security goals are extracted from the protocol’s security requirements. (**b**) The security protocol is put in idealized form. (**c**) Realistic assumptions about the protocol is made. (**d**–**f**) Derivation uses these inputs to derive the stated goals. (**g**) The security goals are checked if the derivation satisfies them. (**h**) Based on the results of (**g**), the protocol is validated as safe or not.

**Figure 3 sensors-21-08383-f003:**
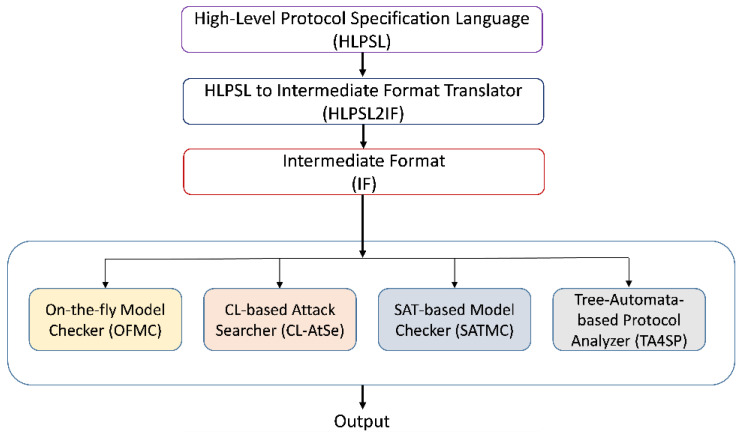
AVISPA system structure.

**Figure 8 sensors-21-08383-f008:**
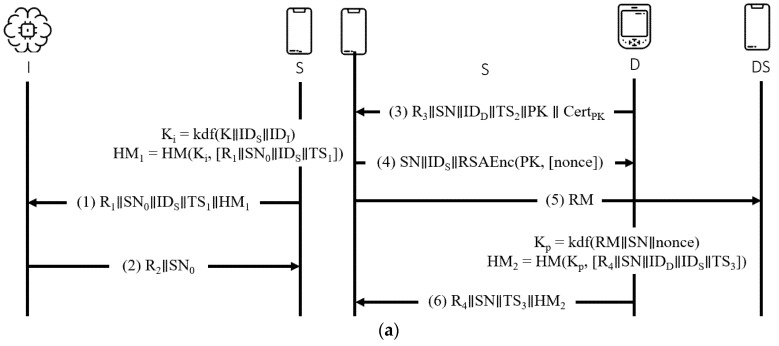
Chi et al.’s protocol. (**a**) Initialization phase. (**b**) Pairing phase. (**c**) Authentication and authorization phase.

**Figure 12 sensors-21-08383-f012:**
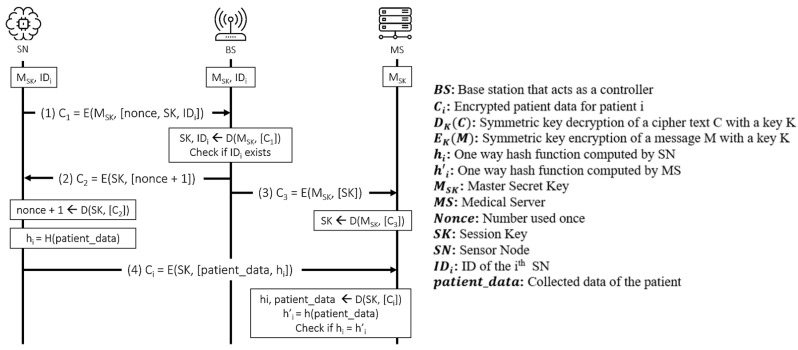
Iqbal et al.’s protocol.

**Figure 17 sensors-21-08383-f017:**
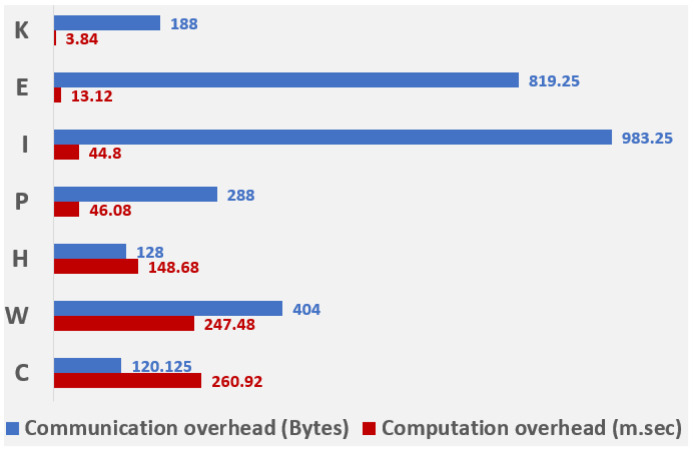
Computational and communication overheads of the authentication protocols.

**Figure 18 sensors-21-08383-f018:**
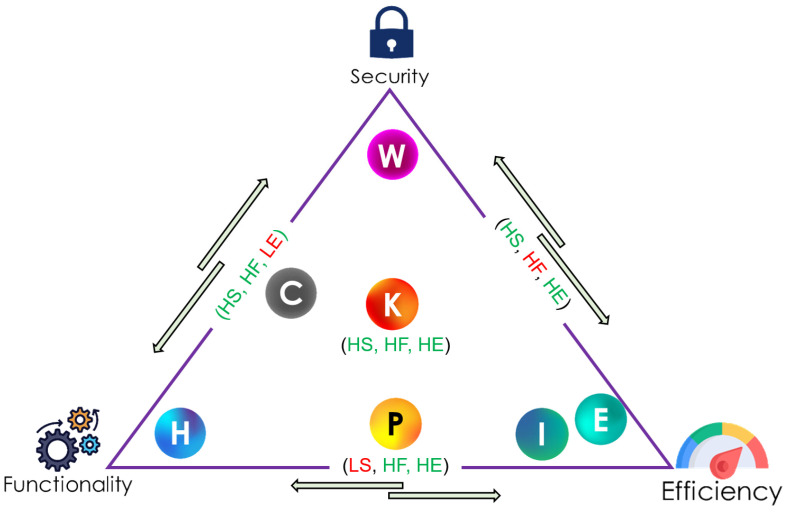
Overall comparison of the authentication protocols using triangular graph.

**Table 1 sensors-21-08383-t001:** BAN logic notations.

Notation	Meaning
M believes U	M believes that the message U is true
M sees U	M receives the message U at any point in time
M said U	M previously sent the message U
M controls U	M has jurisdiction over U
Fresh (U)	U is fresh
$M\overset{S}{↔}N$	S is a secret key shared between M and N
$\overset{S}{\to }M$	S is the M’s public key
$M\overset{S}{\Leftrightarrow }N$	S is a shared secret between M and N.
${\left\{U\right\}}_{K}$	U is encrypted with a key K
U,V	U is combined with V

**Table 2 sensors-21-08383-t002:** BAN logic rules.

Rule Name	Rule
Message Meaning Rule (MM)	$\frac{M\;\mathrm{believes}\;M\overset{S}{↔}N,\;M\;\mathrm{sees}\;{\left\{U\right\}}_{S}\;}{M\;\mathrm{believes}\;N\;\mathrm{said}\;U}$ $\frac{M\;\mathrm{believes}\;M\overset{S}{\Leftrightarrow }N,{M\;\mathrm{sees}\;\langle U\rangle }_{S}\;}{M\;\mathrm{believes}\;N\;\mathrm{said}\;U}$ $\frac{M\;\mathrm{believes}\;\overset{S}{\to }N,M\;\mathrm{sees}\;{\left\{U\right\}}_{L^{-1}}\;}{M\;\mathrm{believes}\;N\;\mathrm{said}\;U}$
Nonce Verification (NV) Rule	$\frac{M\;\mathrm{believes}\;\#\left(U\right),\;M\;\mathrm{believes}\;N\;\mathrm{said}\;U}{M\;\mathrm{believes}\;N\;\mathrm{believes}\;U}$
Jurisdiction (JR) Rule	$\frac{M\;\mathrm{believes}\;N\;\mathrm{controls}\;U,\;M\;\mathrm{believes}\;N\;\mathrm{believes}\;U}{M\;\mathrm{believes}\;U}$
Freshness (FR) Rule	$\frac{M\;\mathrm{believes}\;\mathrm{fresh}\left(U\right)}{M\;\mathrm{believes}\;\mathrm{fresh}\left(U,V\right)}$
Decomposition (DR) Rule	$\frac{M\;\mathrm{sees}\;\left(U,V\right)}{M\;\mathrm{sees}\;U}$
Belief Conjunction (BC) Rule	$\frac{M\;\mathrm{believes}\;U,M\;\mathrm{believes}\;V\;}{M\;\mathrm{believes}\;\left(U,V\right)}$ $\frac{M\;\mathrm{believes}\;N\;\mathrm{believes}\;\left(U,V\right)\;}{M\;\mathrm{believes}\;N\;\mathrm{believes}\;U}$ $\frac{M\;\mathrm{believes}\;N\;\mathrm{said}\;\left(U,V\right)\;}{M\;\mathrm{believes}\;N\;\mathrm{said}\;U}$
Diffie–Hellman (DH) Rule	$\frac{M\;\mathrm{believes}\;N\;\mathrm{said}\;\overset{g^{V}}{\to }N,M\;\mathrm{believes}\;\overset{g^{U}}{\to }M\;}{M\;\mathrm{believes}\;M\overset{g^{\mathrm{UV}}}{↔}\;N}$ $\frac{M\;\mathrm{believes}\;N\;\mathrm{said}\;\overset{g^{V}}{\to }N,M\;\mathrm{believes}\;\overset{g^{U}}{\to }M\;}{M\;\mathrm{believes}\;M\overset{g^{\mathrm{UV}}}{\Leftrightarrow }N}$

**Table 3 sensors-21-08383-t003:** Comparison by security strength.

Notation	INT	CNF	AUT	SKA	PFS	RAP
Khan et. al	✓	✓	✓	✓	✓	✓
Wu et. al.	✓	✓	✓	✓	✓	✓
Chi et. al.	✓	✓	✓	✓	✓	✓
Parvez et al.	✗	∆	✓	✓	✓	✗
Iqbal et. al.	✗	✗	✓	✓	✗	✗
He and Zeadally	∆	✓	✓	✓	✓	✓
Ellouze et. al.	∆	∆	✓	✓	∆	✓

✓ denotes that the scheme supports a particular requirement; ✗ denotes that the scheme does not support a particular requirement, ∆ denotes calls for the additional assumption.

**Table 4 sensors-21-08383-t004:** Comparison by functionality.

Notation	EMA	KUM	ADP	APP	ANO
Khan et. al	✗	✗	A	Generic	✓
Wu et. al.	✗	✗	A	Generic	✗
Chi et. al.	✓	✗	A	Generic	✓
Parvez et al.	✗	✗	A	Generic	✓
Iqbal et. al.	✗	✗	A+	Generic	✗
He and Zeadally	✗	✗	A+	Generic	✓
Ellouze et. al.	✓	✗	A-	Specific	✗

✓ denotes that the scheme supports a particular requirement; ✗ denotes that the scheme does not support a particular requirement. A+: Adaptable for the already implanted device, A: adaptable for yet to be implanted but manufactured, A-: difficult to adapt.

**Table 5 sensors-21-08383-t005:** Approximate computational time in millisecond.

Notation	Meaning	Computational Time
T_H_	Cryptographic hash function	0.32
T_SE_	Symmetric encryption	5.6
T_SD_	Symmetric decryption	5.6
T_EM_	Elliptic curve point multiplication	63
T_AE_	Asymmetric encryption	62
T_AD_	Asymmetric decryption	36
T_SIGN_	RSA-1024 digital signature	7
T_VER_	RSA-1024 digital signature verification	~0
T_XOR_	Bitwise XOR operation	0.32

**Table 6 sensors-21-08383-t006:** Approximate message length in Bits.

Message Type	Message Length
Public key size for an RSA encryption	1024
Length of an RSA Signature	1024
RSA-1024 digital Certificate	602
Elliptic curve point	320
Cryptographic hash function (SHA-1)	160
Key size for AES encryption	128
Identity	32
Timestamp	32
Sequence number	32
Symmetric decryption	32

**Table 7 sensors-21-08383-t007:** Computational overheads of the authentication protocols.

Protocol	Overhead	Time (Milliseconds)
Khan et. al	12T_H_ + 23T_XOR_	3.84
Wu et. al.	9T_H_ + 3T_SE_ +3T_SD_ + 1T_AE_ +1T_AD_ + 2T_SIGN_ + 2T_VER_	247.48
Chi et. al.	16T_H_ + 4T_SE_ + 4T_SD_ + 1T_AE_ +1T_AD_ + 2T_SIGN_ + 2T_V ER_	260.92
Parvez et al.	4T_H_ + 4T_SE_ + 4T_SD_	46.08
Iqbal et. al.	4T_SE_ + 4T_SD_	44.80
He and Zeadally.	4T_H_ + 4T_SE_ + 4T_SD_ + 6T_EM_	148.68
Ellouze et. al.	6T_H_ + 1T_SE_ + 1T_SD_	13.12

**Table 8 sensors-21-08383-t008:** Communication overheads of the authentication protocols.

Protocol	Roundtrips	Overheads (Bits)
Khan et. al	2	1504
Wu et. al.	6	6554
Chi et. al.	9	7866
Parvez et al.	3.5	2304
Iqbal et. al.	2	1024
He and Zeadally.	3.5	3232
Ellouze et. al.	3	961

## Data Availability

Not applicable.

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
