# Peer review of "Can Formal Security Verification Really Be Optional? Scrutinizing the Security of IMD Authentication Protocols"

_sensors, 2021, doi:10.3390/s21248383_

Round 1

Reviewer 1 Report

This paper gives a thorough survey and analysis of IMD security issues, challenges and use one from the variants of modal logic (BAN-Logic) and another from model checking (AVISPA) to perform formal security verification for the authentication schemes proposed to safeguard IMDs. It compared these schemes concerning security strength, computational overhead, latency, and additional features like emergency authentication, adaptiveness, and key update mechanisms.

Overall the paper is written well with solid research contributions. Some minor corrections are required before acceptance, liste below:

In abstract, please give the full-term of acronyms, such as "BAN-Logic 27" and "AVISPA".

On Page 3, please give the full-term of the acronym "EEPROM". Please also check other acronyms in the paper.

On page 4, "Two important issues with this regard are user anonymity and non-tractability [6], [29]...", here should be "traceability"? Please also check it on Page 32 for "non-tractability"

On page 4, "In addition, when illegitimate users temper the data,...", here should be "tamper"?

On page 4, "three security triads (confidentiality and integrity the other two)", what is "the other two"? maybe just add 'Availability' here.

On page 5, "Also, private key encryption is faster". this should be "symmetric key encryption is faster"?

on Page 5, "Device-existence privacy: his privacy requirement challenges", should be "this"?

Author Response

Thank you for your reviews.

We did our best to improve the quality of our paper. 

Reviewer 2 Report

  1. Methodology format needs to be updated. Current format will be confusing for readers. 
  2. Some of the figures are unclear and figures have to be consistent in format.
  3. Authors have do detailed analysis of IMD architecture and solutions to security but authors should still add some more relevant citations in introduction.

Author Response

(The authors gave the same response as above.)

Round 2

Reviewer 2 Report

Authors have successfully updated the manuscript.